

# The first application of a numerically-exact, higher-order sensitivity analysis approach for atmospheric modelling: implementation of the hyperdual-step method in the Community Multiscale Air Quality Model (CMAQ) version 5.3.2

Jiachen Liu[1], Eric Chen[1], Shannon L. Capps[1]

[1]Department of Civil, Architectural & Environmental Engineering, Drexel University, Philadelphia, Pennsylvania, USA

*Correspondence to*: Shannon L. Capps (shannon.capps@drexel.edu)

**Abstract.** Sensitivity analysis in chemical transport models quantifies the response of output variables to changes in input parameters. This information is valuable for researchers in data assimilation and model development. Additionally,
environmental decision-makers depend upon these expected responses of concentrations to emissions when designing and justifying air pollution control strategies. Existing sensitivity analysis methods include the finite-difference method, the direct decoupled method (DDM), the complex variable method, and the adjoint method. These methods are either prone to significant numerical errors when applied to non-linear models with complex components (e.g., finite difference and complex step methods) or difficult to maintain when the original model is updated (e.g., direct decoupled and adjoint methods). Here, we
present the implementation of the hyperdual-step method in the Community Multiscale Air Quality Model (CMAQ) version 5.3.2 as CMAQ-hyd. CMAQ-hyd can be applied to compute numerically exact first- and second-order sensitivities of species concentrations with respect to emissions or concentrations. Compared to CMAQ-DDM and CMAQ-adjoint, CMAQ-hyd is more straightforward to update and maintain while it remains free of numerical errors as those augmented models do. To evaluate the accuracy of the implementation, the sensitivities computed by CMAQ-hyd are compared with those calculated
with other traditional methods or a hybrid of the traditional and advanced methods. We demonstrate the capability of CMAQ-hyd with the newly implemented gas-phase chemistry and biogenic aerosol formation mechanism in CMAQ. We also explored the cross-sensitivity of monoterpene nitrate aerosol formation to its anthropogenic and biogenic precursors to show the additional sensitivity information computed by CMAQ-hyd. Compared with the traditional finite difference method, CMAQ-hyd consumes fewer computational resources when the same sensitivity coefficients are calculated. This novel method
implemented in CMAQ is also computationally competitive with other existing methods and could be further optimised to reduce memory and computational time overheads.




## 1 Introduction

Ambient air pollution, including particulate matter (PM), poses significant threats to human health. According to the Global Burden of Disease study, ambient PM pollution accounted for 4.7 % of the Disability-Adjusted Life Years among all risk factors (Murray et al., 2020) and over 4.1 million deaths (Fuller et al., 2022) in 2019. Therefore, understanding the complex physicochemical and atmospheric transport processes that lead to PM formation is essential to reducing PM and other harmful secondary atmospheric pollutants. Amongst atmospheric scientists, chemical transport models (CTMs) have become essential tools for interpreting observations of and examining inferences about formation processes of atmospheric pollutants. By solving the mass conservation equation for different species based on atmospheric dispersion and transport, photochemical processes, atmospheric chemistry, and aerosol processes, CTMs can provide estimates of primary and secondary air pollutants (Seinfeld and Pandis, 2016). Environmental decision-makers and researchers rely on CTMs to determine appropriate policies to control air pollution and predict atmospheric pollutant concentrations. Experimental studies (e.g., Ng et al., 2008) and measurement campaigns (e.g., Sareen et al., 2016) provide researchers with more insights about the anthropogenic and biogenic aerosol formation processes. These studies ultimately lead to developments and updates to the gas-phase chemistry and aerosol formation mechanisms in CTMs. For these newly added species and mechanisms in CTMs, understanding the sensitivity of aerosol species concentration with respect to their precursor emissions is crucial for determining the priority of primary pollutant emission reductions to achieve atmospheric pollutant reduction objectives.

Sensitivity analysis methods have become invaluable for evaluating uncertainties, understanding concentration-emission relationships in CTMs, and assimilating observations of atmospheric pollutants to improve model parameters. Specifically, the kind of sensitivities here described are the partial derivative of one or more model outputs with respect to one or more model inputs. For instance, if the model has input variables as $X$ and output variables as $Y$, the $n^{th}$-order sensitivity coefficient of one output variable $Y_i$ to one input variable $X_i$ can be represented as the $n^{th}$-order partial derivative of $Y_i$ to $X_i$, $\frac{\partial^n Y_i}{\partial X_i^n}$ (Cohan and Napelenok, 2011). Most sensitivity analysis techniques are formulations of the tangent linear model which provides source-oriented sensitivities or, mathematically, one column of the Jacobian or Hessian at the model state. In contrast, the model adjoint provides receptor-oriented sensitivities or, mathematically, one row of the Jacobian at the model state. Two distinct approaches to developing these models are the continuous approach, in which the derivative of the underlying equation is formulated and then implemented numerically, and the discrete approach, in which the derivative of the numerical solution of the model is formulated (Sandu et al., 2005). Since the model adjoint provides source-oriented sensitivities and is not directly comparable with other methods, including the hyperdual-step method, which is the focus of this work, it will not be further discussed in the following sections. Other augmented model methods including the Integrated Source Apportionment Method (Kwok et al., 2013; Kwok et al., 2015) is based on a different approach thus also not discussed further in the following paragraphs.

The first-order sensitivity coefficient is usually the most useful for CTM applications because it describes the linear relationship between $X_i$ and $Y_i$. Higher-order sensitivities can be helpful when assessing the nonlinear relationships or dynamics



among multiple input variables. Previous studies found that highly nonlinear concentration-emission responses commonly

exist in CTMs, particularly for the formation process of PM (Hakami, 2004; Xu et al., 2018; Tian et al., 2010). Therefore, accurately determining the first-order and second-order relationships are useful for understanding concentration-emission responses in CTMs. Practically, the characteristics of an ideal sensitivity analysis method are numerical accuracy, computational efficiency, and minimal development (Lantoine et al., 2012).

Because analytical sensitivities are impractical for these models, researchers have employed a few numerical methods

to calculate the first- and higher-order sensitivities in CTMs. One such method is the finite difference method (FDM), which is often designated the brute-force method. The FDM is based on the first- or higher-order approximation of the Taylor series expansion from a small perturbation (Boole, 1960). The sensitivities are calculated by running the model multiple times with incrementally different values for the input variables of interest and taking the difference of the resulting concentration fields. While this method is simple to understand and implement, truncation and subtractive cancellation errors can substantially

reduce the accuracy of the calculated sensitivity coefficients, particularly for nonlinear input-output relationships (Fornberg, 1981). Truncation errors originate from neglected higher-order terms in the Taylor series expansion. For instance, suppose a policymaker is interested in calculating the effects of reducing $SO_2$ emission on the total $PM_{2.5}$ concentration. The sensitivity analysis indicates that the first-order partial derivative is positive, and the second-order partial derivative is negative. In that case, only considering the first-order FDM approximation will overestimate the effect of reducing $SO_2$ emission on the total

$PM_{2.5}$ concentration. The truncation error can be minimised by taking a small perturbation step, thus eliminating the impact of higher-order sensitivity terms on the first-order result. However, smaller perturbation steps might lead to subtractive cancellation errors, which stem from the fact that computers cannot distinguish two numbers close to each other. If the perturbation size is within the numerical noise of the model, the numerator difference sometimes approaches zero or the sensitivity information might be meaningless, which causes an inherent tension between reducing the truncation error and

subtractive cancellation error for the FDM. Determining ideal perturbation sizes for different variables is challenging because the ideal perturbation size depends on the input species of interest and other parameters in the model. The necessity of selecting the proper perturbation size for each input variable of interest and running the model multiple times makes the FDM a computationally expensive method to obtain sufficiently accurate sensitivities from CTMs.

As a continuous, source-oriented approach, the decoupled direct method (DDM) eliminates the numerical accuracy

issues of the FDM and improves the computational efficiency of calculating source-oriented sensitivities but only with a hefty development cost. Dunker (1981) introduced to atmospheric modelling the direct method, which involves formulating new sensitivity equations from the original model and solving both sets simultaneously. The direct method has been proven numerically unstable for solving stiff equations, which exist in many chemical transport models (Yang et al., 1997). DDM formulates sensitivity equations like the direct method but separately solves the original and sensitivity equations, which

improves the computational efficiency and stability over the direct method. Dunker et al. (2002) developed DDM-3D and applied it to a three-dimensional air quality model. Hakami et al. (2003) applied extended the method to a higher-order DDM (HDDM) in the gas phase of the Community Multiscale Air Quality Model (CMAQ), while Zhang et al. (2012) augmented



CMAQ-HDDM to include the second-order sensitivities of $PM_{2.5}$ concentration to $NO_x$ and $SO_2$ emissions. Unlike the FDM, the DDM does not incur truncation or subtractive cancellation errors since separate equations are solved for the sensitivities.

The DDM also allows the computation of sensitivities of many outputs to more than one input simultaneously, saving significant computation resources. The major disadvantage of DDM for CMAQ and other CTMs is the difficulty of co-development alongside ongoing scientific model development, which is one purpose of CTMs. The implementation of DDM requires writing sensitivity equations for nonlinear steps, which commonly exist in the chemistry and advection parts of CTMs (Fike and Alonso, 2011). New sensitivity equations must be written when CTMs are updated, reducing the ease of maintenance of DDM in complex CTMs and eliminating the opportunity for sensitivities to be used for evaluation in the process of developing new scientific modules within the CTMs.

The complex variable method (CVM) and the multicomplex step approach (MCX) are the methods most comparable to the hyperdual step method. Lyness and Moler (1967) introduced the concept of using imaginary space to propagate derivatives for functions in real space based on Cauchy integrals. Squire and Trapp (1998) made the idea practical through an elegant truncation of the Taylor series expansion of complex numbers, which allows nearly exact first-order sensitivities if the imaginary perturbation is small enough. Constantin and Barrett (2014) applied CVM on the adjoint method of the global CTM GEOS-Chem to compute near-exact sensitivities with receptor-orientation in one order and source-orientation in the other. Lantoine et al. (2012) developed a multicomplex number system to allow higher-order sensitivities to be calculated to machine precision for functions in real space. These methods require inclusion of a library of overloaded operators to treat the types of numbers required and conversion of the model from real to complex space. The accuracy of these approaches is only limited by ensuring that the imaginary perturbation is small enough, which may require tuning depending on the complexity of the model. CVM does not require as much more memory and computational time as MCX, but both contribute overhead. Both are very easily updated with new scientific modules. The main limitation of CVM and MCX is that sensitivities cannot be propagated through models, like CMAQ, that originally include calculations in imaginary space.

Finally, the method of interest in this work is the hyperdual-step method (HYD), which computes source-oriented first- and second-order sensitivities to machine precision. HYD relies on hyperdual numbers, which are a specific type of generalised complex number developed particularly for first- and second-order derivative calculations (Fike and Alonso, 2011). The HYD, like the CVM or MCX, is an approach based on a Taylor series expansion in a non-real number space. The unique mathematical properties of hyperdual numbers lead to an elegant calculation of first-, second-, and potentially higher-order sensitivities to machine precision without truncation or cancellation errors. Hyperdual numbers have been applied in numerical models in different fields of study to calculate exact first- and second-order derivatives. Cohen and Shoham (2015) applied hyperdual numbers to compute second-order derivatives in multibody kinetics problems. Tanaka et al. (2015) utilized hyperdual numbers to automatically differentiate hyperelastic material models. Rehner and Bauer (2021) applied hyperdual numbers to equation of state modeling and the calculation of critical points. This method, which is applicable to models with calculations in imaginary space, is accurate to machine precision, reasonably computationally expensive, and quite straightforward to update.



Here, we implement the HYD in CMAQ version 5.3.2 to develop the novel augmented model, CMAQ-hyd, and apply it to calculate the sensitivities both inorganic and organic aerosol concentration with respect to their precursor emissions. To our best knowledge, this work represents the first implementation of hyperdual numbers to calculate first- and second-order sensitivities in a CTM. In Section 2, hyperdual numbers and the hyperdual-step method are introduced as well as the process of implementing and evaluating HYD in CMAQ. In Section 3, the evaluation of first- and second-order sensitivities from CMAQ-hyd is conducted, including the computational costs. In Section 4, CMAQ-hyd is applied to understand the influences of anthropogenic and biogenic emissions on select secondary organic aerosol (SOA) concentrations. This work provides an accurate and easily manageable method to compute first- and second-order sensitivities implemented in CMAQ version 5.3.2 and an example of potential application in other complex models where the sensitivities are of interest.

## 2 Methods

### 2.1 Hyperdual numbers and the hyperdual-step method

A hyperdual number (Fike and Alonso, 2011) has four components and is characterised by

$$H = a_0 + a_1\epsilon_1 + a_2\epsilon_2 + a_{12}\epsilon_{12} \qquad (1)$$

where $a_0$, $a_1$, $a_2$, and $a_{12}$ are real numbers, and $\epsilon_1$, $\epsilon_2$, and $\epsilon_{12}$ are non-real parts. The three crucial properties which enable numerically exact first- and second-order sensitivity calculations are:

$$\epsilon_1^2 = \epsilon_2^2 = \epsilon_{12}^2 = 0 \qquad (2)$$

$$\epsilon_1 \neq \epsilon_2 \neq \epsilon_{12} \neq 0 \qquad (2)$$

$$\epsilon_1\epsilon_2 = \epsilon_{12} \qquad (3)$$

The squares of the non-real individual parts equal zero (Eq. 2). The non-real parts themselves do not equal anything in real space (Eq. 3). The multiplication of $\epsilon_1$ and $\epsilon_2$ is equal to the third non-real component $\epsilon_{12}$ (Eq. 4). The addition and multiplication of hyperdual numbers are commutative, and the definitions help eliminate the higher-order terms in a Taylor series expansion. A demonstration of several basic operations is provided in the SI while a more detailed discussion of the mathematical properties of hyperdual numbers is given by Fike and Alonso (2011).

The method of ascertaining sensitivities through a perturbation in non-real space is based on multiplying a hyperdual number with unity in $a_0$ and unity in one of $a_1$, $a_2$, and $a_{12}$ with the independent variable of interest. After model execution, a Taylor series expansion is applied to extract sensitivities. The hyperdual-step method is applied to a scalar function $f(x)$ by multiplying $x$ by the hyperdual number $H_h = 1.0 + h_1\epsilon_1 + h_2\epsilon_2$ and results in

$$f(xH_h) = f(x) + (xh_1\epsilon_1 + xh_2\epsilon_2)f'(x) + \frac{1}{2!}(xh_1\epsilon_1 + xh_2\epsilon_2)^2f''(x) + \frac{1}{3!}(xh_1\epsilon_1 + xh_2\epsilon_2)^3f'''(x)$$
$$+ \cdots \qquad (5)$$





where "…" represents higher order terms in the series. Eliminating all terms that are zero due to the definition of hyperdual numbers (Eq. 2),

$$f(xH_h) = f(x) + (xh_1\epsilon_1 + xh_2\epsilon_2)f'(x) + x^2h_1h_2\epsilon_{12}f''(x) \tag{6}$$

The properties of hyperdual numbers (Eqs. 2–4) lead to two significant results. First, all terms in the Taylor series expansion with derivatives higher than second-order become zero because all values include either $\epsilon_1^2$, $\epsilon_2^2$, or $\epsilon_{12}^2$. Second, the real component is unchanged. Finally, the first- and second-order derivatives are separated into different parts of the hyperdual

number. The first-order derivative exists in either the $\epsilon_1$ or the $\epsilon_2$ term, while the second-order derivatives only exist in the $\epsilon_{12}$ term. The first and second-order derivatives are,

$$f'(x) = \frac{\epsilon_1 part[f(xH_h)]}{xh_1} = \frac{\epsilon_2 part[f(xH_h)]}{xh_2} \tag{7}$$

$$f''(x) = \frac{\epsilon_{12} part[f(xH_h)]}{x^2h_1h_2} \tag{8}$$

where $\epsilon_1 part[]$, $\epsilon_2 part[]$, and $\epsilon_{12} part[]$ represent functions that extract the $a_1$, $a_2$, or $a_{12}$, respectively. Since the derivative computation process (Eqs. 6–9) does not involve subtractions or higher-order sensitivities, the first- and second-order sensitivities calculated by the hyperdual-step method are free from subtractive cancellation and truncation errors. This method

(Eqs. 8–9) extends to vector operations to compute arrays of numerically exact derivatives. For instance, the partial first- and second-order derivatives for $f(\boldsymbol{x})$, where $\boldsymbol{x} = [x_1, x_2, ..., x_n]$, with respect to $x_1$ through a hyperdual-step perturbation to $x_1$ is:

$$\frac{\partial f(\boldsymbol{x})}{\partial x_1} = \frac{\epsilon_1 part[f(\boldsymbol{x}H_{h,x_1})]}{x_1 h_1} = \frac{\epsilon_2 part[f(\boldsymbol{x}H_{h,x_1})]}{x_1 h_2} \tag{9}$$

$$\frac{\partial^2 f(\boldsymbol{x})}{\partial x_1^2} = \frac{\epsilon_{12} part[f(\boldsymbol{x}H_{h,x_1})]}{x_1^2 h_1 h_2} \tag{10}$$

Similarly, two different independent variables $x_1$ and $x_2$ may be perturbed simultaneously. In this case, two arrays of first-order sensitivity and one array of cross-sensitivity result as:


$$\frac{\partial f(\boldsymbol{x})}{\partial x_1} = \frac{\epsilon_1 part[f(\boldsymbol{x}) * H_h]}{x_1 h_1} \tag{11}$$

$$\frac{\partial f(\boldsymbol{x})}{\partial x_2} = \frac{\epsilon_2 part[f(\boldsymbol{x}) * H_h]}{x_2 h_2} \tag{12}$$

$$\frac{\partial^2 f(\boldsymbol{x})}{\partial x_1 \partial x_2} = \frac{\epsilon_{12} part[f(\boldsymbol{x}) * H_h]}{x_1 x_2 h_1 h_2} \tag{13}$$

Therefore, the two variations of the hyperdual-step method will generate either one or two arrays of first-order sensitivities and one array of second-order or cross sensitivities with a single run of the model.



## 2.2 Community Multiscale Air Quality Model and the implementation of the hyperdual-step method

The Community Multiscale Air Quality model (CMAQ) developed by the US EPA is an Eulerian CTMs which can predict air pollutant concentrations on regional and hemispheric scales (Byun and Schere, 2006). CMAQ represents advection, diffusion, gas-phase chemistry, aerosol processes, cloud processes, and photolysis. CMAQ has been applied to predict pollutant concentrations in the atmosphere (Liu et al., 2010; Sayeed et al., 2021), understand fundamental atmospheric chemistry and aerosol formation mechanisms (Zhu et al., 2018; Li et al., 2019), and guide policy-making processes (Chemel et al., 2014; Che

et al., 2011; Li et al., 2019; Ring et al., 2018). CMAQ is used in the regulatory process of the US EPA when states, tribes, or local jurisdictions demonstrate how they will attain the National Ambient Air Quality Standard (NAAQS) and or comply with the Regional Haze Rule (Mebust et al., 2003). CMAQ solves the atmospheric diffusion equation shown in Eq. (14) to calculate the concentrations of gaseous and aerosol species in the atmosphere.

$$\frac{\partial c_i}{\partial t} = -\nabla(\boldsymbol{u}c_i) + \nabla(\boldsymbol{K}\nabla c_i) + R_i + E_i \tag{14}$$

where $c_i$, $\boldsymbol{u}$, $\mathbf{K}$, $R_i$ and $E_i$ are the concentration of species $i$, the wind velocity vector, the diffusivity tensor, the change in

concentration due to chemical reaction of species $i$, and the emissions rate of species $i$, respectively. Species concentrations are stored in a multidimensional array and propagated through different scientific modules within the model. For this work, CMAQ was run with 12 km by 12 km horizontal resolution with 35 vertical layers, 100 columns, and 80 rows over the Southeast US on July 1st, 2016, GMT (U.S. Environmental Protection Agency, 2019) . The gas-phase chemistry mechanism used is Carbon Bond 6 (Luecken et al., 2019).

In CMAQ, the hyperdual-step method was implemented by strategically converting the model to use hyperdual numbers (Figure 1). First, the operators were overloaded by translating a C-based library from Fike and Alonso (2011) to Fortran ("HDMod") and augmenting it to treat multidimensional data as required by CMAQ. HDMod defines a hyperdual version of all possible calculations related to the chemical concentration array was developed. The library includes basic arithmetic operations, such as addition and subtraction, as well as more advanced functions like trigonometric functions. Before

being applied to CMAQ, the operator overloading library was separately tested using a testing framework developed by Pellegrini and Russell (2016). Secondly, the CMAQ variable containing species concentration information and all other variables that depend on it were converted from real numbers to the newly defined hyperdual number type. The source code was carefully analysed to select only the necessary variables for conversion. Many variables in CMAQ do not need to be altered because they do not influence the main concentration array. This highly detailed process helped minimise the additional

computational burden of the model since calculations with hyperdual numbers are more computationally intensive than those with real numbers. For instance, one hyperdual multiplication operation shown in Eq. (5) results in five more additions and nine more multiplications than an operation with real numbers. According to Fike and Alonso (2011), the computational cost of a hyperdual calculation ranges from 4 to 14 times higher than the original operation. Applying hyperdual numbers to all the variables in a CFD model results in approximately 10 times higher computational cost (Fike and Alonso, 2011). Thirdly, the



first- and second-order sensitivities of the species concentrations to perturbed emissions are included in the new hyperdual array, which is then saved to additional output files using the same structure as the output of the original concentration array. As a result, first- and second-order sensitivities can be propagated through the model without significantly modifying the source code. The modification efforts mainly focused on determining the variables that must be converted to the hyperdual type. Consequently, updating CMAQ-hyd when there are changes to the original model is a simple process that involves

converting only the newly added variables to the hyperdual type. This simplicity is an advantage over other computational techniques, such as the DDM and adjoint method, which compute numerically exact sensitivities but require more complex and time-consuming update procedures, including writing new equations.

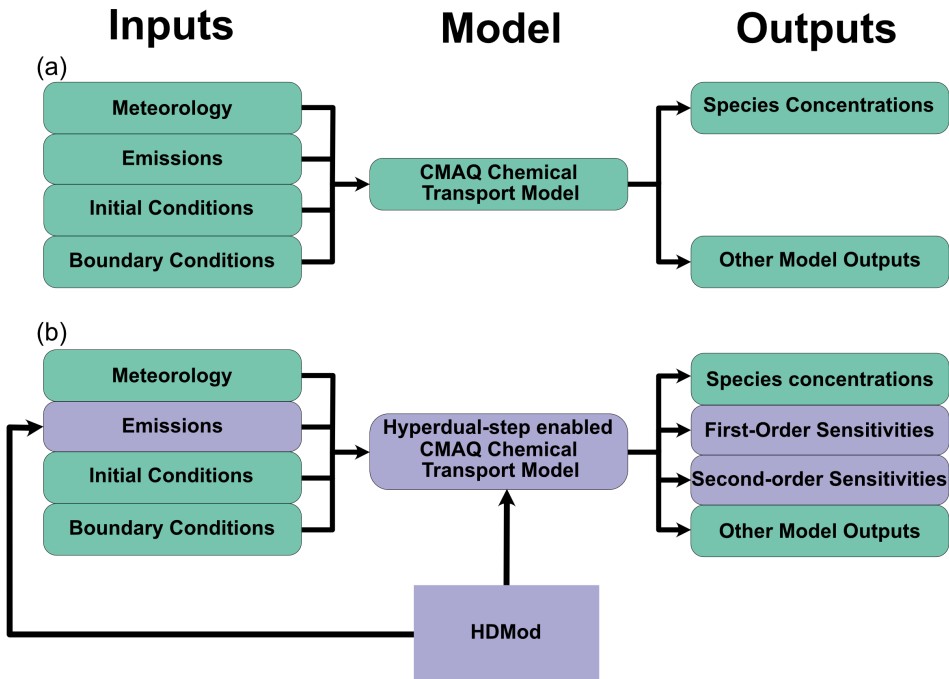

**Fig 1.** A schematic showing the original CMAQ model compared with the CMAQ-hyd. **(a)** The original CMAQ model (green). **(b)** The CMAQ-hyd model. The CMAQ-hyd incorporates a hyperdual number overloading modules. The purple components in (b) represent changes made to the original CMAQ model. Changes were made to the modules and emission processing parts of CMAQ. The first- and second-order sensitivities of species concentrations with respect to selected precursor concentrations or emissions are new outputs of the model.

Some source code alternations were made to overcome the numerical instabilities related to hyperdual calculations in CMAQ's aerosol module. CMAQ uses the thermodynamic module ISORROPIA to compute the partitioning of inorganic

aerosol and gas-phase species (Fountoukis and Nenes, 2007; Nenes et al., 1998). ISORROPIA can run in either the forward or reverse mode. The forward mode of ISORROPIA takes the sum of gas and aerosol species concentrations, along with the relative humidity and temperature, to determine the partitioning of species in either the gas or aerosol phase. The reverse mode of ISORROPIA starts with known concentrations of aerosol species and can output the species concentration in the solid,



liquid, and gas phases. CMAQ uses the reverse mode to compute the thermodynamics of coarse-mode inorganic aerosols. The
reverse-mode solution leads to unrealistic sensitivities calculated by the HYD when the aerosol pH is close to neutral. One
previous study found that the reverse ISORROPIA fails to capture the actual behaviour of inorganic aerosol when the pH is
close to 7 (Hennigan et al., 2015), and the necessary change was to ignore the sensitivity calculations in CMAQ when the pH
of coarse mode aerosol is close to neutral. The original implementation was numerically unstable during iterations for upper-
layer cells with low temperature and pressure in the forward-mode inorganic aerosol computation for Aitken-mode and fine-
mode aerosols. In the original CMAQ model, ISORROPIA is run in the forward mode without limiting the temperature and
pressure of the simulation. The determination process of species concentrations involves an iterative method which sometimes
is numerically unstable during iterations for upper layer cells with low temperature and pressure. The forward-mode
ISORROPIA is only called when the cell temperature exceeds 260 K, and cell pressure exceeds 20,000 Pa to increase the
numerical stability of the model. A similar set of temperature and pressure limits was applied to the adjoint of CMAQ (Zhao
et al., 2020). These changes do not affect the species concentrations computed by CMAQ and ensure that the sensitivity
computation process is stable. For the simplicity of development, we applied a Fortran 90-compliant version of ISORROPIA
to CMAQ-hyd.

**2.3 Evaluating sensitivities from CMAQ-hyd**

CMAQ-hyd produces sensitivities that can be semi- or fully-normalized for concentrations from any range of grid
cells and times with respect to emissions or concentrations from any range of grid cells and times. Here, we consider the semi-
normalised sensitivities of output concentrations on the ground layer to input emissions averaged over time, which would be
a typical application for an environmental decision maker. First-order semi-normalized sensitivities, $s_{NO_x}^{PM_{2.5}}$, and second-order
semi-normalised sensitivities, $s_{NO_x}^{(2)PM_{2.5}}$, of PM2.5 concentrations, $C_{PM_{2.5}}$, to NOx (NO+NO2) emissions, $E_{NO_x}$, exemplify
sensitivities relevant to environmental decision makers (Eqs. 16–17).

$$s_{NO_x}^{PM_{2.5}} = \frac{\partial C_{PM_{2.5}}}{\partial E_{NO_x}} E_{NO_x} \qquad (16)$$

$$s_{NO_x}^{(2)PM_{2.5}} = \frac{\partial^2 C_{PM_{2.5}}}{\partial E_{NO_x}^2} E_{NO_x}^2 \qquad (17)$$

Semi-normalised sensitivities reduce the complexity of interpretation by providing sensitivities in the units of the
concentration per percent change of emissions. Here, $C_{PM_{2.5}}$ and $E_{NO_x}$ are the concentration of PM2.5 and emission of NOx at a
given cell in the modelling domain each averaged in time, respectively. The semi-normalised sensitivities also scale down the
impact from cells with low emission rates, which is consistent with the concentration reduction that is realistic. Similarly, the
time-averaged, semi-normalised cross-sensitivity of PM2.5 to both NOx and monoterpene is denoted as $s_{NO_x,TERP}^{(2)PM_{2.5}}$, with $E_{TERP}$
representing the emission of monoterpenes (Eq. 18).



$$s_{NO_x,TERP}^{(2)PM_{2.5}} = \frac{\partial^2 C_{PM_{2.5}}}{\partial E_{NO_x} \partial E_{TERP}} (E_{NO_x} E_{TERP}) \tag{18}$$

The evaluation of CMAQ-hyd in first order is done by performing a comparison of the sensitivities calculated by the hyperdual-step method against those from the FDM (Eq. 19). The comparison is illustrated with an example of calculating cell-specific sensitivities of PM$_{2.5}$ concentration to NO$_x$ emissions. The first-order sensitivity of PM$_{2.5}$ concentration at the end of a 24-hour simulation to cumulative NO$_x$ emission perturbation is given by

$$s_{NO_x}^{PM_{2.5},t=24} \approx \frac{C_{PM_{2.5},c,r,l,t=24}^{inc} - C_{PM_{2.5},c,r,l,t=24}^{dec}}{\sum_{t=0}^{t=24} E_{NO_x,c,r,l=0,t}^{inc} - \sum_{t=0}^{t=24} E_{NO_x,c,r,l=0,t}^{dec}} \sum_{t=0}^{t=24} E_{NO_x,c,r,l=0,t}^{orig} \tag{19}$$

where the subscripts $c$, $r$, and $l$ represent the column, row, and layer; the subscript $t$ represents the time from the start of the model run; and the superscripts $inc$, $dec$, and $orig$ represent the initial perturbation direction. For instance, $C_{PM_{2.5},c,r,l,t=24}^{inc}$ is the concentration of PM$_{2.5}$ for each column, row, and layer at 24 hours into the run, with an increase in NO$_x$ emissions throughout the model run. Unless otherwise noted, the relative perturbation size for first-order FDM calculations is 125 % and 75 % for domain-wide emissions. The average ground-layer sensitivities for the 24-hr simulation time are computed. Previous studies have found smaller perturbation sizes for inorganic aerosol sensitivity calculations in CMAQ using FDM are more prone to numerical noise (Zhang et al., 2012). The semi-normalised sensitivity of each cell is computed with the central difference method and is only an approximation of the actual sensitivities due to subtractive cancellation and truncation errors. The numerator is the difference between PM$_{2.5}$ concentrations with persistent increases or decreases in NO$_x$ emissions. The denominator is the total emission perturbation of NO$_x$ emission. The sensitivities are semi-normalised by the sum of NO$_x$ emissions in the base model run. The calculated first-order semi-normalised sensitivities will have units of µg m$^{-3}$. Sensitivities calculated with this method (Eq. 19) can only be an approximation due to numerical errors mentioned in Section 2.1.

The semi-normalised sensitivity of PM$_{2.5}$ concentrations with respect to NO$_x$ emissions using a hyperdual perturbation of $H_a = 1 + a_1\epsilon_1 + a_2\epsilon_2$ is computed by the hyperdual-step method as

$$s_{NO_x}^{PM_{2.5},t=24} = \frac{\epsilon_1 part[C_{PM_{2.5},c,r,l,t=24}^{orig}]}{a_1} = \frac{\epsilon_2 part[C_{PM_{2.5},c,r,l,t=24}^{orig}]}{a_2} \tag{19}$$

The first-order semi-normalised sensitivity can be computed with either the $\epsilon_1$ or the $\epsilon_2$ part. The $\epsilon_1$ or the $\epsilon_2$ part of the PM$_{2.5}$ concentration divided by the initial perturbation in the $\epsilon_1$ or $\epsilon_2$ space, respectively. The emissions in the denominator will cancel out with the semi-normalised emissions.

Although the FDM can be applied to compute second-order sensitivities in CMAQ, previous studies have shown that the results are noisy and highly dependent on the perturbation sizes (Zhao et al., 2020; Zhang et al., 2012). The second-order sensitivity evaluation is between a hybrid hyperdual-finite-difference method (HYD-FDM) and the hyperdual-step method. The hybrid sensitivity calculation is given by:



$$s_{NO_x}^{(2)PM_{2.5}} \approx \frac{\dfrac{\epsilon_1 part[C_{PM_{2.5},c,r,l,t=24}^{inc}]}{a_1 \sum_{t=0}^{t=24} E_{NO_x,c,r,l=0,t}^{inc}} - \dfrac{\epsilon_1 part[C_{PM_{2.5},c,r,l,t=24}^{dec}]}{b_1 \sum_{t=0}^{t=24} E_{NO_x,c,r,l=0,t}^{dec}}}{\sum_{t=0}^{t=24} E_{NO_x,c,r,l=0,t}^{inc} - \sum_{t=0}^{t=24} E_{NO_x,c,r,l=0,t}^{dec}} \left(\sum_{t=0}^{t=24} E_{NO_x,c,r,l=0,t}^{orig}\right)^2 \qquad (20)$$

where two separate simulations were run: one with increased and another with decreased initial $NO_x$ emissions. The perturbation on emissions for two runs is $H_a = 1 + a_1 \epsilon_1 + a_2 \epsilon_2$ for the run with increased initial $NO_x$ emission, and $H_b = 1 + b_1 \epsilon_1 + b_2 \epsilon_2$ for the run with decreased initial emission of $NO_x$. The HYD-FDM uses the regular finite difference on the difference between first-order sensitivities calculated by using the $\epsilon_1$ part of the hyperdual-step results. The sensitivity in this equation is an estimate and subject to numerical errors because it includes the usage of FDM.

The second-order sensitivity calculated by the hyperdual-step method is shown in Eq. (21) below.

$$s_{NO_x}^{(2)PM_{2.5}} = \frac{\epsilon_{12} part[C_{PM_{2.5},c,r,l,t=24}^{orig}]}{a_1 a_2} \qquad (21)$$

The hyperdual-step method uses the $\epsilon_{12}$ part of the output variable and divide it by the multiplication of $a_1$ and $a_2$. The second-order sensitivities calculated only by the HYD method are numerically exact. All the sensitivities are computed for each cell, and comparisons between the finite difference and the hyperdual-step method are performed on a cell-to-cell basis.

## 3 Results and Discussion

### 3.1 Evaluation of the first- and second-order sensitivities

We evaluated the implementation of CMAQ-hyd by comparing the first-order sensitivities of various species in CMAQ calculated by HYD with a hyperdual-step perturbation described in Section 2.3 (HYD sensitivities) and FDM with a 50% domain-wide emission perturbation (FD sensitivities). Overall, different HYD and FD sensitivities agree well, as evidenced by the close alignment of the points on the blue identity line, which represents perfect agreement, in most panels of Figure 2. The slope and $R^2$ values for all comparisons are provided in Table 1, and additional slope and $R^2$ values are provided





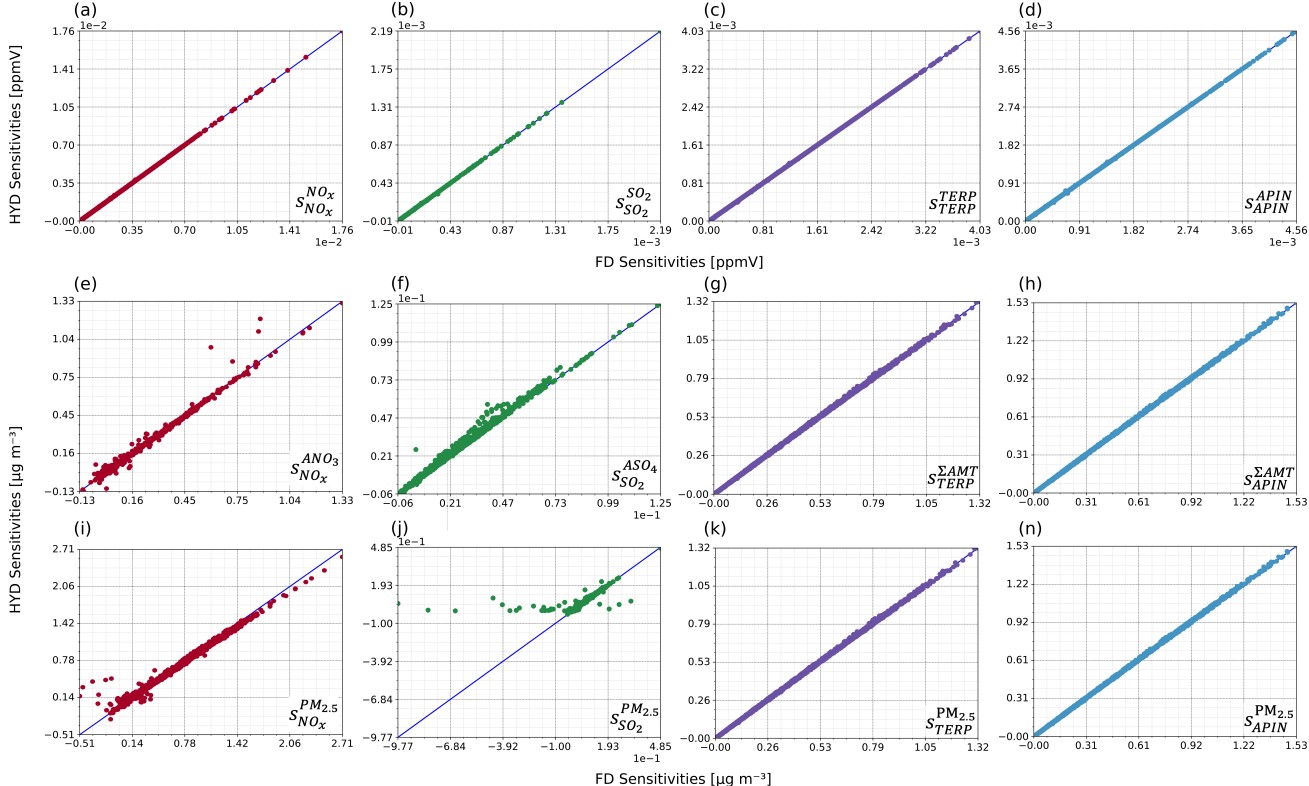

**Figure 2.** The comparisons of first-order sensitivities on the ground layer calculated by the hyperdual-step method (HYD sensitivities) and FDM (FD sensitivities). The sensitivities are color-coded by the perturbed emissions (i.e., $NO_x$, $SO_2$, TERP, and APIN). (a)-(d) are the gas-phase species sensitivities with respect to their emissions. APIN denotes α-pinene and TERP denotes all other monoterpene species. (e)-(h) are examples of aerosol phase products with respect to their precursors. $ANO_3$ denotes the total aerosol phase nitrate products. $ASO_4$ denotes the total aerosol sulphate products. ΣAMT denotes the total aerosol photooxidation products from monoterpene. (i)-(n) are the total $PM_{2.5}$ concentration with respect to gas-phase precursors. The sensitivities calculated are noted at the bottom-right corner of each panel with the notation pattern mentioned in Section 2.3.

in Table S1 and Table S2. Specifically, the slopes and $R^2$ of gas-phase species concentration on the ground layer with respect

to their emissions on the ground layer (Figs. 2a–2d) are all 1.00 (Table 1), indicating minimal nonlinearity in these

relationships, as expected.

Secondary aerosol formation is a more nonlinear process, which is explored by using inorganic or organic aerosol

concentrations with respect to select precursors (Figs. 2e-2h). Nonlinearities in the modelled processes lead to discrepancies

between HYD and FD sensitivities without tuning the FD sensitivity to capture the slope about the model state more exactly.

The slopes and $R^2$ values of the trendline between these HYD and FD sensitivities range from 0.99 to 1.00 and 1.00 to 1.04

(Table 1), respectively. The comparisons between HYD and FD sensitivities of $s_{NO_x}^{ANO_3}$ and $s_{SO_2}^{ASO_4}$ show slight deviations from

the identity line, indicating some nonlinearity in their formation processes (Fig. 2e and Fig. 2f). Most points representing the

HYD and FD sensitivities of total monoterpene photooxidation products to monoterpenes ($s_{TERP}^{AMTNO_3}$) and alpha-pinene

($s_{APIN}^{AMTNO_3}$) remain on the identity line (Fig. 2g and Fig. 2h).





**Table 1.** Slope and $R^2$ of the comparisons of first-order sensitivities of ground layer species concentrations to domain-wide perturbations. The plots are shown in Figure 2.

| First-order sensitivities: slope, $R^2$ | | | |
|---|---|---|---|
| NOₓ | SO₂ | TERP | APIN |
| $s_{NO_x}^{NO_x}$: 1.00, 1.00 | $s_{SO_2}^{SO_2}$: 1.00, 1.00 | $s_{TERP}^{TERP}$: 1.01, 1.00 | $s_{APIN}^{APIN}$: 1.00, 1.00 |
| $s_{NO_x}^{ANO_3}$: 1.00, 0.99 | $s_{SO_2}^{ASO_4}$: 1.04, 1.00 | $s_{TERP}^{\Sigma AMT}$: 1.01, 1.00 | $s_{APIN}^{\Sigma AMT}$: 1.01, 1.00 |
| $s_{NO_x}^{PM_{2.5}}$: 0.96, 0.99 | $s_{SO_2}^{PM_{2.5}}$: 0.65, 0.63 | $s_{TERP}^{PM_{2.5}}$: 1.01, 1.00 | $s_{APIN}^{PM_{2.5}}$: 1.03, 1.00 |

305       Regulatory models are often used to evaluate the response of total PM₂.₅ to emissions changes, so the sensitivities of total PM₂.₅ concentration to the four different precursor emissions are evaluated (Figs. 2i–2n). The HYD and FD sensitivities of $s_{NO_x}^{PM_{2.5}}$, $s_{TERP}^{PM_{2.5}}$, and $s_{APIN}^{PM_{2.5}}$ (Fig. 2i, Fig. 2k, and Fig. 2n) agree well, with slope and $R^2$ values ranging from 0.96 to 1.03 and 0.99 to 1.00, respectively (Table 1). However, the agreement between HYD and FD sensitivities of $s_{SO_2}^{PM_{2.5}}$ (Fig. 2j) is much lower, with a slope of 0.65 and an $R^2$ value of 0.63 (Table 1). Notably, although $s_{SO_2}^{PM_{2.5}}$ is usually positive, as evidenced by

most of the points on the identity line, the $s_{SO_2}^{PM_{2.5}}$ calculated by FDM sensitivities have a few negative values where the HYD and FD sensitivities disagree. Because it is highly unlikely that an 50% increase in SO₂ emission leads to a decrease in PM₂.₅ concentration, the negative FD sensitivities likely arise from truncation errors inherent to the FDM since the perturbation sizes

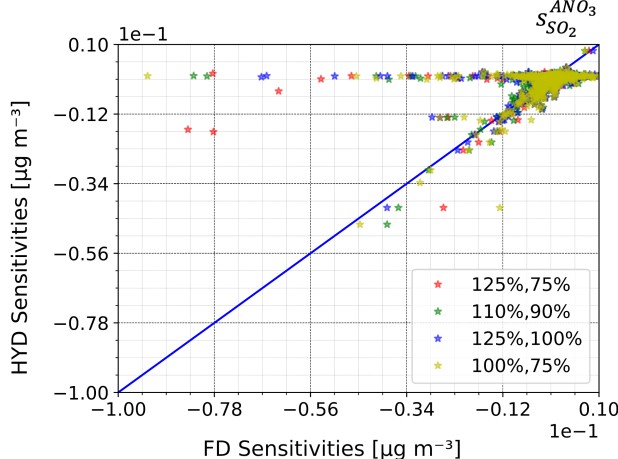

**Figure 3.** The comparisons of first-order sensitivities of ground layer aerosol nitrate (ANO₃) concentration with respect to domain-wide perturbation of SO₂ emission. The HYD sensitivities are on the y-axis, and the FDM sensitivities are on the x-axis. The different perturbation sizes of FDM are color-coded. For instance, the red stars represent a central difference calculation with increased and decreased 25% of domain-wide SO₂ emission.





are large (i.e., 50% emissions perturbation). Though it is possible to refine the perturbation size to one more suitable for this particular relationship of emissions to concentration as demonstrated in the next section, this difference in one of twelve

comparisons shows one of the strengths of HYD, which is the irrelevance of the perturbation size to the exactness of the resulting sensitivity.

To further illustrate the impact of nonlinear relationships between emissions and concentrations on FD sensitivities, we explored the sensitivity of ground-level aerosol nitrate to emissions of sulphur dioxide, $s_{SO_2}^{ANO_3}$, calculated with different

perturbation sizes using the FDM. Our analysis revealed a low level of agreement between the FD and HYD sensitivities in the base case scenario, where the domain-wide $SO_2$ emission was perturbed by 125% and 75%, with a slope of 0.10 and $R^2$ of 0.30 (Table S1). The FD sensitivities with the base case

perturbation (125 %, 75 %) and three other perturbation size pairs (110 %, 90 %; 125 %, 100 %; 100 %, 75 %) are shown in Fig. 3. The FDM sensitivities calculated with different perturbation sizes are colour-coded and plotted against the HYD sensitivities. While many points with different FDM

perturbation sizes aligned closely on the identity line, indicating a first-order inverse relationship between aerosol phase nitrate and $SO_2$ emissions, some positive sensitivities deviated from the identity line. These positive FD sensitivities likely resulted from numerical errors inherent to the FDM,

regardless of different perturbation sizes. The lack of overlapping points among the sensitivities calculated by FDM with different perturbation sizes (Fig. 3) suggests that the FD sensitivities heavily depend on the perturbation sizes. This result also shows the low credibility of FD sensitivities,

particularly for highly nonlinear relationships where the truncation errors could be large. Our findings demonstrate the importance of using other methods, including the HYD, which are not prone to truncation or cancellation errors for

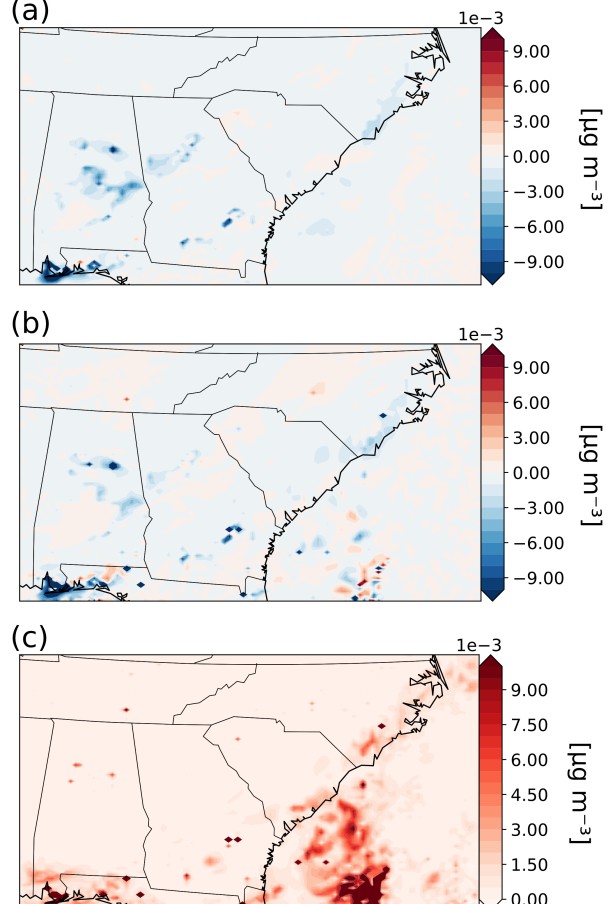

**Figure 4.** Comparisons of the sensitivities of aerosol phase nitrate ($ANO_3$) with respect to $SO_2$ emission calculated by HYD and FDM on a map. a) The HYD sensitivities. b) The average of the FDM sensitivities with four different perturbation sizes (125%, 75%; 110%, 90%; 125%, 100%; 100%, 75%). c) The range of the FDM sensitivities with four different perturbation sizes (125%, 75%; 110%, 90%; 125%, 100%; 100%, 75%).

nonlinear relationships in CTMs. We also compared the spatial distribution of HYD sensitivities (Fig. 4a) against the average

(Fig. 4b) and the range (Fig. 4c) of the FD sensitivities with four different perturbation sizes. Maps of FDM sensitivities with four different perturbation sizes are available in Figure S1. Differences are evident between the HYD and the average FD





sensitivities in central North Carolina and Tennessee as well as off the coasts of Georgia and South Carolina. The HYD predicts slightly negative sensitivities in North Carolina and Tennessee while the FDM predicts slightly positive values. The average FDM sensitivities off the coast of Georgia and South Carolina were noisy, with alternating positive and negative sensitivities,

while the HYD sensitivities were much less noisy. In addition, the range of FDM sensitivities with different perturbation sizes was large (Fig. 4c), especially off the coast Georgia and South Carolina. The results shown in Fig. 4b and Fig. 4c illustrate the dependence of FDM sensitivities on perturbation sizes especially for highly nonlinear relationships.

We also compared the second-order HYD sensitivities with those calculated from the hybrid HYD-FDM method (hybrid second-order sensitivities) described in Section 2.3 using one-to-one plots with identity lines for each panel (Fig. 4)

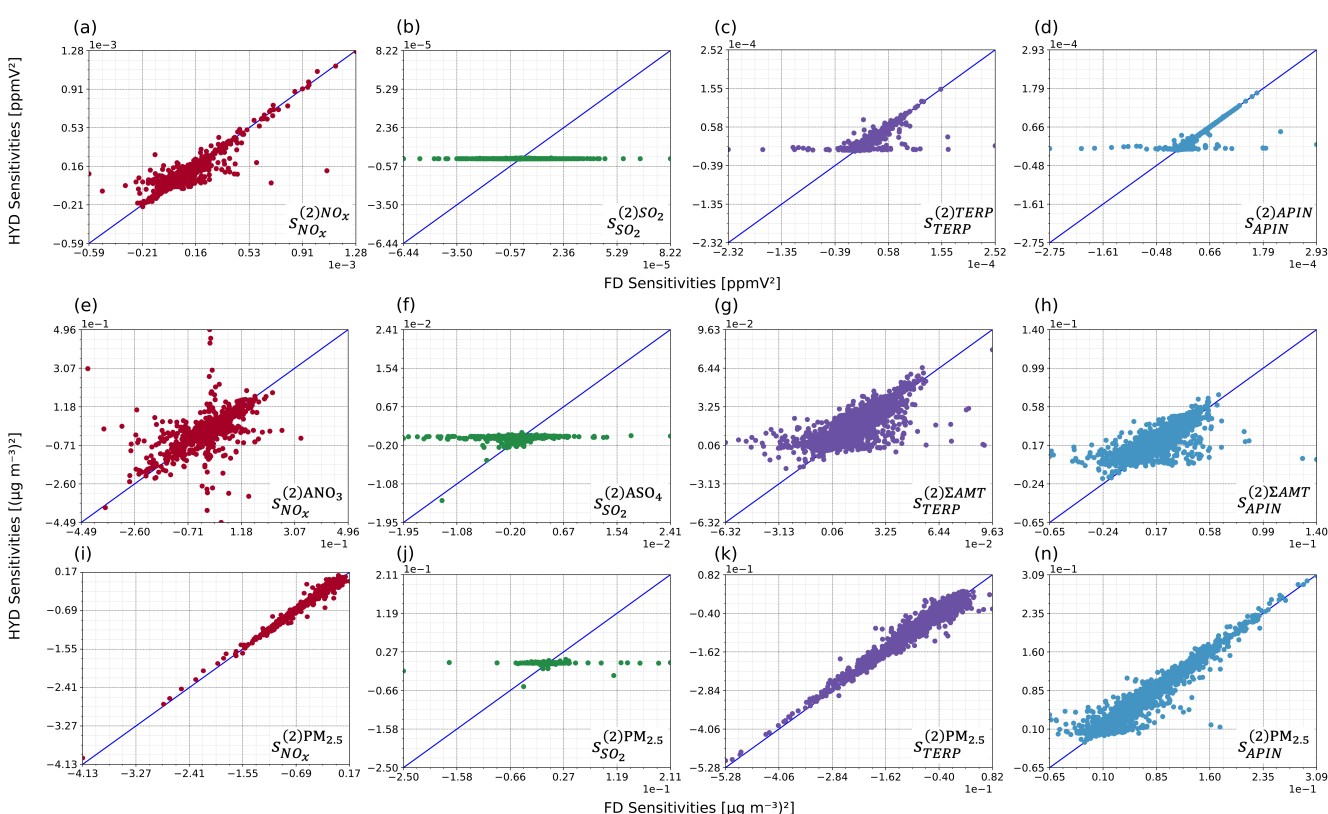

**Figure 5.** The comparisons of second-order sensitivities on the ground layer calculated by the hyperdual-step method (HYD sensitivities) and FDM (FD sensitivities). The sensitivities are color-coded by the perturbed emissions (i.e., $NO_x$, $SO_2$, TERP, and APIN). (a)-(d) are the gas-phase species sensitivities with respect to their emissions. APIN denotes $\alpha$-pinene and TERP denotes all other monoterpene species. (e)-(h) are examples of aerosol phase products with respect to their precursors. $ANO_3$ denotes the total aerosol phase nitrate products. $ASO_4$ denotes the total aerosol sulphate products. $\Sigma AMT$ denotes the total aerosol photooxidation products from monoterpene. (i)-(n) are the total $PM_{2.5}$ concentration with respect to gas-phase precursors. The sensitivities calculated are noted at the bottom-right corner of each panel with the notation pattern mentioned in section 2.3

along with the slope and $R^2$ values (Table 2). Additional slopes and $R^2$ values for second-order sensitivities can be found in Table S1 and Table S2. Overall, the agreement between HYD and the hybrid second-order sensitivities is good, except for those to $SO_2$ emissions. This result can be attributed to the numerical errors in the first-order sensitivities to $SO_2$, as illustrated



in Fig. 2j and Fig. 3. Computing second-order sensitivities with the hybrid method, which includes FDM, is expected to add numerical noise. Except for $s_{SO_2}^{(2)SO_2}$, the gas phase species concentrations to their emissions still exhibit good agreement, with

slopes and $R^2$ values ranging from 0.84 to 0.85 and 0.84 to 0.86, respectively (Table 2). The hybrid second-order sensitivities are sometimes large, while HYD predicts close to zero sensitivities. This result is especially evident in $s_{TERP}^{(2)TERP}$ (Fig. 5c) and $s_{APIN}^{(2)APIN}$ (Fig. 5d). This spread in the hybrid sensitivities likely originates from the FDM step, which is subject to numerical errors. Figures 5e–5h show the HYD and HYD-FDM sensitivities of aerosol phase product concentrations to the precursor emissions for this modelling period. Except for $s_{SO_2}^{ASO_4}$, the slope and $R^2$ values range from 0.61 to 0.82 and 0.38 to 0.71,

respectively (Table 2). The degree of agreement for $s_{NO_x}^{ANO_3}$ is slightly lower than those for $s_{TERP}^{\Sigma AMT}$ and $s_{APIN}^{\Sigma AMT}$, indicating more nonlinearity in the formation process from $NO_x$ to aerosol nitrate. The second-order sensitivities of total $PM_{2.5}$ to different precursors demonstrate excellent agreement with slope and $R^2$ values ranging from 0.95 to 0.99 and 0.96 to 0.99 (Table 2), again excluding the one to $SO_2$. The second-order sensitivities of $PM_{2.5}$ to $NO_x$ and α-pinene are primarily negative, while positive to monoterpenes. These findings have important implications for the formation process of $PM_{2.5}$ from monoterpenes

and α-pinene, which will be discussed in detail in the next section.

**Table 2.** Slope and $R^2$ of the comparisons of second-order sensitivities of ground layer species concentrations to domain-wide perturbations of selected emissions. The plots are shown in Figure 5.

| Second-order sensitivities: slope, $R^2$ | | | |
|---|---|---|---|
| $NO_x$ | $SO_2$ | TERP | APIN |
| $s_{NO_x}^{NO_x}$: 0.84, 0.84 | $s_{SO_2}^{SO_2}$: 0.00, 0.00 | $s_{TERP}^{TERP}$: 0.84, 0.84 | $s_{APIN}^{APIN}$: 0.85, 0.86 |
| $s_{NO_x}^{ANO_3}$: 0.61, 0.38 | $s_{SO_2}^{ASO_4}$: 0.04, 0.06 | $s_{TERP}^{\Sigma AMT}$: 0.79, 0.71 | $s_{APIN}^{\Sigma AMT}$: 0.82, 0.71 |
| $s_{NO_x}^{PM_{2.5}}$: 0.98, 0.99 | $s_{SO_2}^{PM_{2.5}}$: 0.01, 0.01 | $s_{TERP}^{PM_{2.5}}$: 0.95, 0.98 | $s_{APIN}^{PM_{2.5}}$: 0.99, 0.96 |


## 3.2 Sensitivities of biogenic aerosol formation in the southeast US computed by CMAQ-hyd

In this section, the first- and second-order sensitivities of several biogenic aerosols to both anthropogenic and biogenic aerosol precursors in the southeast US are explored. The importance of calculating second-order sensitivities is demonstrated through the spatial distributions of the first- and second-order sensitivities of total aerosol phase monoterpene photooxidation

product (ΣAMT) and $PM_{2.5}$ concentrations (Fig. 6). The first-order sensitivities (Figs. 6a–6d) are predominantly positive, indicating that an increase in either TERP or APIN emissions will lead to an increase in ground-layer ΣAMT and $PM_{2.5}$





concentrations. While $s_{TERP}^{\Sigma AMT}$ (Fig. 6a) and $s_{APIN}^{\Sigma AMT}$ (Fig. 6b) have similar values, $s_{TERP}^{PM_{2.5}}$ (Fig. 6c) is slightly larger than $s_{APIN}^{PM_{2.5}}$ (Fig. 6d) due to the formation of other species, such as aerosol phase monoterpene nitrate products (AMTNO$_3$).

The second-order sensitivities (Fig. 6e–6h) provide additional information about how ΣAMT and PM$_{2.5}$ concentrations respond to changes in APIN and TERP emissions. The $s_{TERP}^{(2)\Sigma AMT}$ (Fig. 6e) and $s_{APIN}^{(2)\Sigma AMT}$ (Fig. 6f) are generally positive, indicating the concentration of ΣAMT to monoterpene and α-pinene emissions is convex. An increase in either monoterpene or α-pinene emissions will lead to an increase in $s_{TERP}^{\Sigma AMT}$ and $s_{APIN}^{\Sigma AMT}$. If we only consider first-order sensitivities, the effect of changes in TERP or APIN emissions on ΣAMT concentrations would be underestimated. On the other hand, the $s_{TERP}^{(2)PM_{2.5}}$ (Fig. 6g) is mostly positive, while $s_{APIN}^{(2)PM_{2.5}}$ (Fig. 6h) is mostly negative. The distinct behaviour of second-order sensitivities of PM$_{2.5}$ concentration to either TERP or APIN emissions exemplify the importance of considering second-order sensitivities for these nonlinear formation processes. Only considering the first-order sensitivities often leads to overestimating or underestimating the effects. The accurate second-order sensitivity information can help researchers understand the relationships of concentration to emissions more thoroughly and develop emission control strategies for specific aerosol precursor emissions.

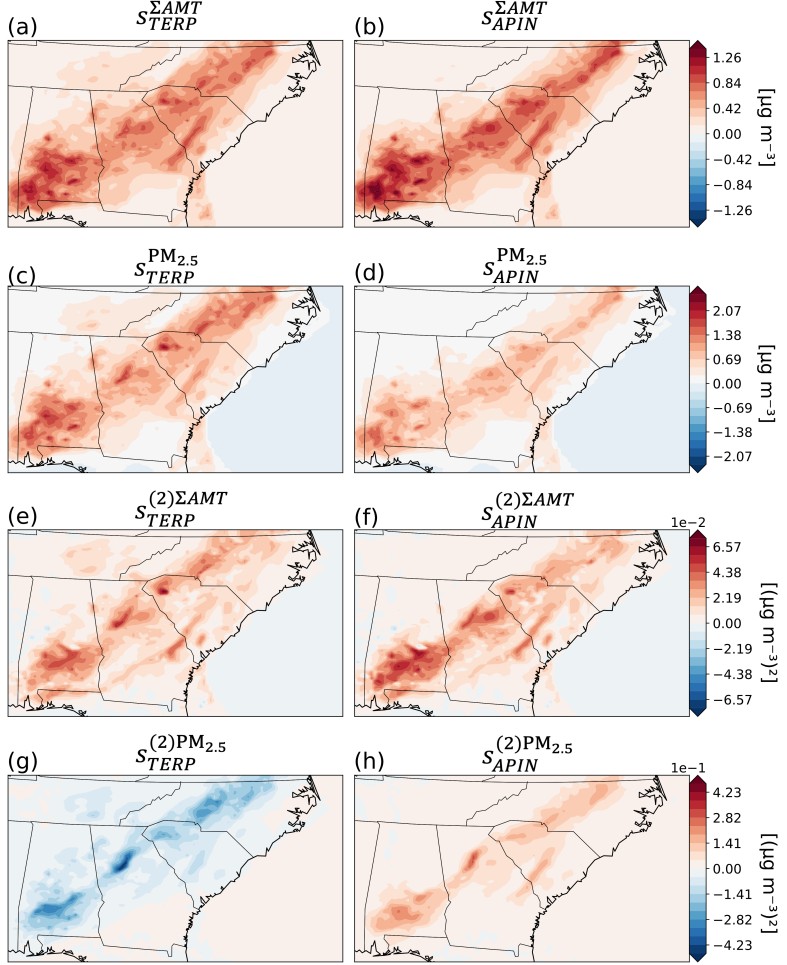

**Figure 6.** The first- and second-order sensitivities of total aerosol monoterpene photooxidation products (ΣAMT) and PM$_{2.5}$ with respect to monoterpenes (TERP) and alpha-pinene (APIN) emissions plotted on a map. (a)-(d) are the first-order sensitivities and (e)-(h) are the second-order sensitivities.





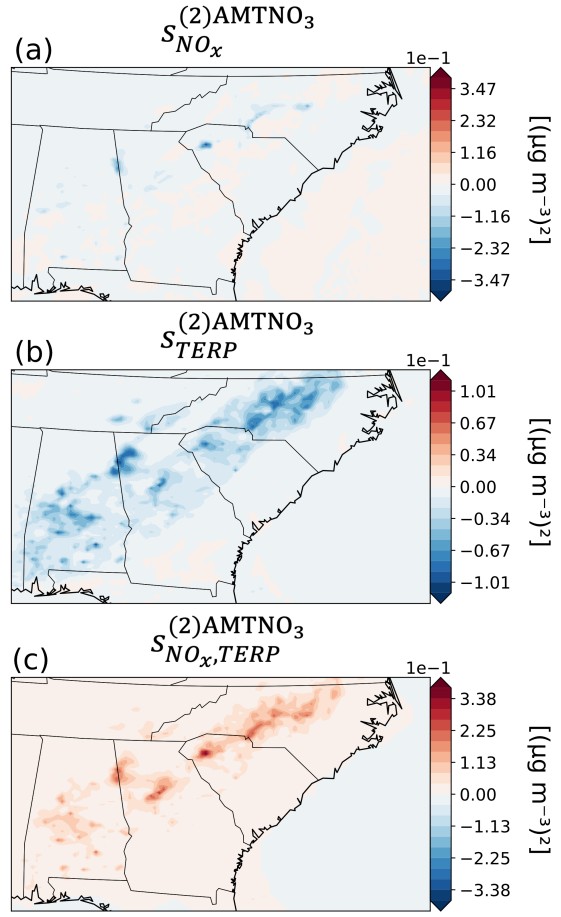

**Figure 7.** The second-order sensitivities and cross-sensitivities of aerosol phase monoterpene nitrate products (AMTNO₃) with respect to NO$_x$ and monoterpenes (TERP) emissions plotted on a map. (a). second-order sensitivities of AMTNO₃ with respect to NOx. (b). second-order sensitivities of AMTNO₃ with respect to TERP. (c) cross-sensitivities of AMTNO₃ with respect to NO$_x$ and TERP

We used the formation of aerosol monoterpene nitrate, AMTNO₃, as an example of the importance of computing the cross-sensitivity, especially for complex anthropogenic-biogenic aerosol formation processes. The formation of AMTNO₃ is influenced primarily by two precursors: NO$_x$ and monoterpenes. NO$_x$ is primarily emitted anthropogenically while monoterpenes primarily originate from biogenic sources. The first- and second-order sensitivities of AMTNO₃ to NO$_x$ or TERP can help researchers and environmental decision makers estimate the nonlinear effects of emissions changes on concentrations of secondary pollutants. The cross-sensitivity of AMTNO₃ with respect to both NO$_x$ and TERP emissions, $s_{NO_x,TERP}^{(2)AMTNO_3}$, is a valuable tool for answering complex research questions. For instance, researchers can use $s_{NO_x,TERP}^{(2)AMTNO_3}$ to predict how an increase in monoterpene emissions would affect the first-order sensitivities of AMTNO₃ to NO$_x$. Since biogenic emissions of monoterpenes are temperature-dependent, understanding how anthropogenic emissions of NO$_x$ will affect AMTNO₃ formation with changing terpene emissions in future scenarios is crucial for designing effective air pollution control strategies. Computing the cross-sensitivity is especially challenging with traditional methods since determining the proper perturbation for two species using FDM is even harder than calculating second-order sensitivities with FDM. The distinct values of $s_{NO_x}^{(2)AMTNO_3}$, $s_{TERP}^{(2)AMTNO_3}$, and $s_{NO_x,TERP}^{(2)AMTNO_3}$ demonstrate the value of the HYD method (Fig. 7). The spatial distributions of $s_{NO_x}^{AMTNO_3}$ and $s_{TERP}^{AMTNO_3}$ are included in Fig. S2. Overall, the second-order sensitivities are negative over land in the southeast US. The $s_{NO_x}^{(2)AMTNO_3}$ is generally smaller than $s_{TERP}^{(2)AMTNO_3}$, indicating the

relationship between AMTNO₃ and TERP emissions is more nonlinear than that between AMTNO₃ and NO$_x$. The cross-sensitivities $s_{NO_x,TERP}^{(2)AMTNO_3}$ are mostly positive over the southeast US. Based on the cross-sensitivity results, we can conclude that an increase in TERP emission will make the first-order sensitivity of AMTNO₃ to NO$_x$ larger. A warmer climate in the future would likely increase the impact of anthropogenic NO$_x$ on AMTNO₃ concentration in the atmosphere. This kind of information





is invaluable for researchers and environmental decisionmakers to evaluate complex secondary organic aerosol formation with
multiple anthropogenic and biogenic precursors.

**3.3 Computational cost of the CMAQ-hyd**

The practical application of any sensitivity analysis in CTMs depends heavily upon its computational cost. Previous works using operator overloading approaches resulted in high computational costs due to additional mathematical operations,
making this approach computationally unfavourable compared to other existing methods. For instance, the implementation of CVM in GEOS-CHEM is 4.5 times higher than the regular model (Constantin and Barrett, 2014). To evaluate the computational efficiency of CMAQ-hyd, we compared it with a standard CMAQ model using different computational resources on a supercomputing cluster. The total wall time of an identical run of the original CMAQ model and CMAQ-hyd are shown for the when using different numbers of nodes (Fig. 8). The CMAQ-hyd and regular CMAQ runs were performed
with 1, 2, 4, and 8 nodes on the supercomputing cluster. Profiling of the model was completed at the level of the scientific modules, special subroutines, or other important components of CMAQ. The scientific processes are Chem (gas-phase chemistry), Aero (aerosol dynamics and thermodynamics), Vdiff (vertical diffusion), Hadv (Horizontal advection), Phot (Photolysis), Cldproc (cloud processes), Hdiff (horizontal diffusion), and Zadv (vertical advection). The MPI_Barrier is a special subroutine used for synchronising processes among parallel processors after vertical diffusion and before all three other
transport processes. Other processes necessary for CMAQ, including the I/O processes, are included in the "Other" category. The details of the high-performance computing cluster used can be found in the SI.



With the same computing resources, the total computation time of the CMAQ-hyd is approximately 2.5 (2.44–2.56) times longer. Despite the additional computation burden, CMAQ-hyd remains computationally competitive with the traditional FDM when calculating derivatives. One run of CMAQ-hyd generates the same amount of first- and second-order sensitivity

information as at least three runs of regular CMAQ. The relatively low computational cost of CMAQ-hyd, compared to the

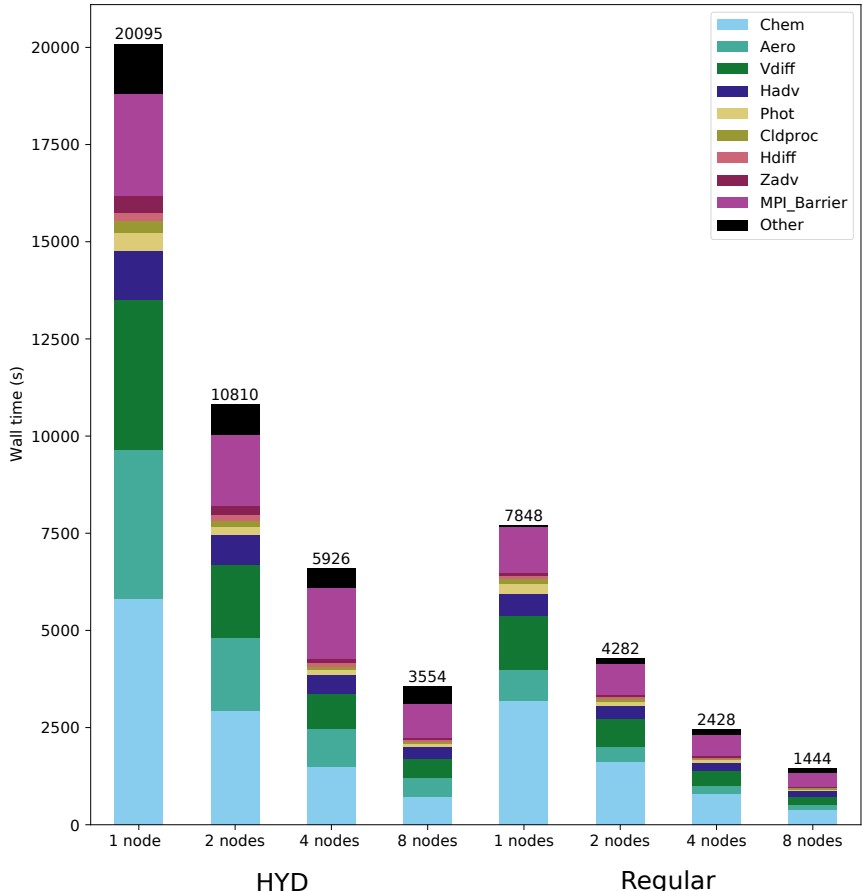

**Figure 8.** Computational cost of CMAQ-hyd separated by different modules. The CMAQ-hyd and regular CMAQ simulations with different number of 1, 2, 4, and 8 nodes and are shown on the x-axis. The wall time is shown on the y-axis.

previous operator overloading approach, may be due to the selective modification of the source code. In contrast to GEOS-CHEM CVM (Constantin and Barrett, 2014), only parts of the model that involve calculating the main species concentration array use hyperdual calculations. The computation time of CMAQ-hyd also scales well with increases in computational resources, similar to the original CMAQ. The overall memory overhead of the CMAQ-hyd is approximately 25 GB for this

simulation. A parallel input/output (I/O) approach may be applied to reduce the possibility of potential memory overflow in processor 0 (Wong et al., 2015). The computation time of each module is detailed in Table S3.





Chem, Aero, and Vdiff are the most computationally expensive modules in both CMAQ and CMAQ-hyd. The relative computational cost of Aero is higher in the CMAQ-hyd than in the regular CMAQ. Future work can potentially reduce the computational cost by ignoring sensitivity propagations during the iterative root-finding process in select subroutines, since only the output concentrations from these subroutines are used in the later part of the model. This is also a significant advantage of any operator overloading-based approach (Fike and Alonso, 2011). The I/O process of newly added first- and second-order sensitivity output files increases the computational cost; however, the I/O of species concentration files has a much lower computational cost than other computing modules in CMAQ for this specific scenario.

## 4 Conclusion

We demonstrate the implementation of the hyperdual-step method in CMAQ version 5.3.2 to formulate CMAQ-hyd. This novel model retains the majority of CMAQ code with slight modifications in declarations of selected variables and the addition of sensitivity computation modules. The novel model can be applied to compute exact first-order, second-order, and cross-sensitivities of pollutant concentrations efficiently and accurately to precursor emissions with a single model run. Compared with traditional sensitivity analysis methods, CMAQ-hyd is computationally competitive with conventional methods and easier to maintain than other existing advanced methods (DDM and adjoint). The development process of CMAQ-hyd is also more straightforward than that of other advanced methods.

We developed and validated the hyperdual-step module "HDMod", which involves mathematical operations. This module can also be applied to other numerical models where first- and second-order sensitivities are of interest. We further validated the development of CMAQ-hyd against the FDM and FDM-HYD hybrid method to ensure the correctness of the implementation. During the validation process, CMAQ-hyd demonstrated the ability to compute sensitivities free from numerical noise, different from those calculated by the FDM. HDMod can potentially be applied to other numerical models written in Fortran to produce first- and second-order sensitivities.

The computation of second-order sensitivities is crucial for researchers and environmental decision-makers to decide the priority and extent of controls of specific types of emissions to reduce atmospheric pollutant concentrations. For instance, the second-order sensitivity of $PM_{2.5}$ concentration to monoterpenes and α-pinene provided additional information about relationships of emissions to concentrations in CMAQ. With the additional second-order sensitivity information, the curvature of the concentration responses to emissions changes improves the estimate of how a specific pollutant concentration would respond to changes in emissions. The simplicity of computing cross-sensitivities with CMAQ-hyd is another advantage of this augmented model. Cross-sensitivities are especially useful in nonlinear processes with two precursors. The synergistic effect of anthropogenic and biogenic emission on aerosol concentrations (e.g., $NO_x$ and monoterpene on $AMTNO_3$) is essential for researchers to predict the dynamics between two potential pollutants and for environmental decision-makers to propose policy implementations under different climate scenarios in the future.



Although CMAQ-hyd remains computationally competitive with the traditional finite-difference method, it is still
computationally intensive and has a memory overhead. We plan to implement optimisations for iterative processes in CMAQ
and apply the parallel I/O approach to reduce the memory overhead on the compute node where all the information is gathered.
The implementation of checkpointing of sensitivities after specific subroutines is also a potential advantage of CMAQ-hyd
and will provide valuable information of how each module or even each line of the model affects the sensitivities, akin to a
process analysis approach. This checkpointing feature cannot be easily implemented with other methods such as FDM, DDM,
and the adjoint method.

In conclusion, we have developed and evaluated CMAQ-hyd, a novel augmented model to compute first-order,
second-order, and cross-sensitivities free from numerical noise in CMAQ. Our successful implementation also provides an
example of the hyperdual-step method that may be applicable for other CTMs where sensitivities are helpful.


*Code and Data Availability.* CMAQv5.3.2 is available at https://github.com/USEPA/CMAQ/tree/5.3.2 and is archived at
https://doi.org/10.5281/zenodo.4081737 (U.S. Environmental Protection Agency, 2020). The CMAQ-hyd model is archived
at https://doi.org/10.5281/zenodo.7938726 (Liu, 2023). Both the CMAQv5.3.2 and the CMAQ-hyd models are under MIT
licenses. The input data for the simulation experiments is available at https://doi.org/10.15139/S3/IQVABD (U.S.
Environmental Protection Agency, 2019).

*Author contribution.* JL developed the model code and performed the simulations with direction from SC. EC helped develop
and test the "HDMod" library. JL prepared the manuscript with help from SC.


*Competing Interests.* The authors declare that they have no conflict of interest.

*Acknowledgements.* The work is supported by National Science Foundation CAREER Award Grant No. 1944669. The authors
would also like to thank Dr. Ryan P. Russell for kindly providing the testing framework of multicomplex numbers, which
inspired the development of the testing framework for hyperdual numbers.





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
