# Peer review of "The first application of a numerically-exact, higher-order sensitivity analysis approach for atmospheric modelling: implementation of the hyperdual-step method in the Community Multiscale Air Quality Model (CMAQ) version 5.3.2"

_EGUsphere, 2023_

## Author Comment (AC1)

**Response to RC1**

Jiachen Liu[1], Eric Chen[1], Shannon L. Capps[1]

[1]Department of Civil, Architectural & Environmental Engineering, Drexel University, Philadelphia, Pennsylvania, USA

*Correspondence to*: Shannon L. Capps (shannon.capps@drexel.edu)

*For the convenience of the reviewer, we have included the comments from Reviewer 1 in black, normal font style and our responses indented in green, normal font style. The changes made to the manuscript or supplemental information corresponding to our response are provided in green, bold style font with the context of line numbers from the original manuscript and the original, remaining text in the normal font style.*

In this paper, Liu et al., have applied the hyper-dual sensitivity analysis approach to a chemical transport model (CMAQ). They find the method to be both accurate and computationally relatively efficient for calculating first and second order sensitivities. In general, the manuscript is scientifically sound and well written.

- First and foremost, we would like to thank the reviewer for giving careful attention to this work and for the favorable assessment of it.
- We have also corrected a miscellaneous inconsistent spelling of modeling from **lines 128 to 129**:

  Lines 128 to 129: Rehner and Bauer (2021) applied hyperdual numbers to equation of state **modelling** and the calculation of critical points.

"DDM-3D" was developed by Yang et al., in 1997.

- Thank you for the correction. We have revised the **lines 93 to 96** describing the development of DDM-3D in the manuscript.

  Lines 93 to 96: **On the other hand,** DDM formulates sensitivity equations like the direct method but separately solves the original and sensitivity equations. This **approach** improves the computational efficiency and stability **compared to** the direct method. **Yang et al. (1997) was the first to apply the DDM-3D method in a three-dimensional chemical transport model.**

The development of (5) needs to be further explained. Given it is a Taylor series expansion, about what value is the expansion?

- Thank you for highlighting the need for more background on applying the hyperdual step method. We have made the connection to the real-valued Taylor series expansion more explicit in the text and have directed the reader to the SI, where a derivation of the multiplication of a real number and hyperdual perturbation leading to Eq. (5) is now included.

  The first- and second-order sensitivities are in the $\epsilon_1$, $\epsilon_2$, and $\epsilon_{12}$ terms. After further simplification, Eq. (5) becomes Eq. (6), and the numerically exact sensitivities are separated and listed in Eq. (7) and Eq. (8). We have also included a sentence describing Eq. (S7).

Lines 151-159: **Akin to the Taylor series expansion about the real value of $x$ in the finite difference method**, the method of ascertaining sensitivities through a perturbation in **hyperdual space is based on a Taylor series expansion in an orthogonal dimension of the number**. **Specifically**, a hyperdual number with unity in $a_0$ and unity in one of $a_1$, $a_2$, and $a_{12}$ **is multiplied** with the independent variable of interest. After model execution, a Taylor series expansion is applied to extract sensitivities. **For instance,** the hyperdual-step method is applied to a scalar function $f(x)$ by multiplying $x$ by the hyperdual number $H_h = 1.0 + h_1\epsilon_1 + h_2\epsilon_2$, **which** results in:

$$f(xH_h) = f(x) + (xh_1\epsilon_1 + xh_2\epsilon_2)f'(x)$$
$$+ \frac{1}{2!}(xh_1\epsilon_1 + xh_2\epsilon_2)^2 f''(x) \tag{5}$$
$$+ \frac{1}{3!}(xh_1\epsilon_1 + xh_2\epsilon_2)^3 f'''(x) + \cdots$$

where "…" represents higher order terms in the series. Eliminating all terms that are zero due to the definition of hyperdual numbers (Eq. 2) **leads to**

$$f(xH_h) = f(x) + (xh_1\epsilon_1 + xh_2\epsilon_2)f'(x) + x^2 h_1 h_2 \epsilon_{12} f''(x) \tag{6}$$

**where $f(xH_h)$ is a hyperdual number**.

The properties of hyperdual numbers (Eqs. 2–4) lead to two significant results. First, all terms in the Taylor series expansion with derivatives higher than second-order become zero because all values include $\epsilon_1^2$, $\epsilon_2^2$, or $\epsilon_{12}^2$. Second, the real component is unchanged. **A more detailed expansion of terms can be found in Eq. S7 in the SI or the original development of hyperdual numbers, following the multiplication rule between a hyperdual and a real number** (Fike and Alonso, 2011).

Line 15, SI:

$$f(xH_h) = f\big(x * (1.0 + h_1\epsilon_1 + h_2\epsilon_2)\big)$$
$$= f(x + xh_1\epsilon_1 + xh_2\epsilon_2)$$
$$= f(x) + (xh_1\epsilon_1 + xh_2\epsilon_2)f'(x) + \frac{1}{2!}(xh_1\epsilon_1 + xh_2\epsilon_2)^2 f''(x) + \frac{1}{3!}(xh_1\epsilon_1 + xh_2\epsilon_2)^3 f'''(x) + \cdots$$
$$= f(x) + (xh_1\epsilon_1 + xh_2\epsilon_2)f'(x) + \frac{1}{2}\big(x^2 h_1^2 \epsilon_1^2 + 2x^2 h_1\epsilon_1 h_2\epsilon_2 + x^2 h_2^2 \epsilon_2^2\big)f''(x)$$
$$= f(x) + xh_1\epsilon_1 f'(x) + xh_2\epsilon_2 f'(x) + x^2 h_1 h_2 \epsilon_{12} f''(x)$$

**The Taylor expansion of the multiplicative hyperdual perturbation (Eq. 5) is shown in Eq. S7.**

In (16) and (17) is $E_{NOx}$ a function of space or time? If yes, the derivatives calculated are very complex, and indeed, it would be good for the authors to explain exactly how they are taking those derivatives and how the set of mathematical operations are being done. If $E_{NOx}$ is not space or time dependent, what is it (how is it mathematically defined)? I think I know what they are trying to do, but the current representation needs to be clarified and made mathematically more precise. They should indicate the spatial and temporal dependencies in the variables.

- Thank you for this question and helpful comment. In responding to it, we noticed and corrected some errant numbering for the equations in the original manuscript. There were

two occurrences of Equation (2) in the original manuscript from **lines 145 to 150**. We have revised the manuscript to correct this mistake.

- We have also corrected the numbering of Eqs. 15–19 from **lines 252 to 281**, which were mistakenly labeled as Eqs. 16–19, with two occurrences of Eq. 19. We have added to now Eqs. 15-17 the subscripts that formerly were only introduced in what is now Eq. 18.

Lines 236-250: Here, **for the sake of illustration**, we consider the semi-normalised sensitivities of **time-averaged** output concentrations on the ground layer, **l=0**, to input emissions averaged over time, **t,** for any given cell as indicated by the column, **c,** and row, **r.** First-order semi-normalized sensitivities, $s_{NO_x}^{PM_{2.5}}$, and second-order semi-normalised sensitivities, $s_{NO_x}^{(2)PM_{2.5}}$, of **ground-level** PM$_{2.5}$ concentrations, $C_{PM_{2.5},c,r,l=0,t}$, to NO$_x$ (NO+NO$_2$) emissions, $E_{NO_x,c,r,l,t}$, exemplify sensitivities relevant to environmental decision makers (Eqs. **15–16**).

$$s_{NO_x}^{PM_{2.5}} = \frac{\partial \overline{C_{PM_{2.5},c,r,l=0,tPM_{2.5}}}|_t}{\partial E_{NO_x,c,r,l,t}|_t} \overline{E_{NO_x,c,r,l,t}}|_t \tag{15}$$

$$s_{NO_x}^{(2)PM_{2.5}} = \frac{\partial^2 \overline{C_{PM_{2.5},c,r,l=0,tPM_{2.5}}}|_t}{\partial E_{NO_x,c,r,l,t}^2|_t} \overline{E_{NO_x,c,r,l,t}^2}|_t \tag{16}$$

Semi-normalised sensitivities reduce the complexity of interpretation by providing sensitivities in the units of the concentration per percent change of emissions. The semi-normalised sensitivities also scale down the impact from cells with low emission rates, which is consistent with the concentration reduction that is realistic **to expect**. Similarly, the time-averaged, semi-normalised cross-sensitivity of PM$_{2.5}$ to both NO$_x$ and monoterpene is denoted as $s_{NO_x,TERP}^{(2)PM_{2.5}}$, with $E_{TERP}$ representing the emission of monoterpenes (Eq. **17**).

$$s_{NO_x,TERP}^{(2)PM_{2.5}} = \frac{\partial^2 \overline{C_{PM_{2.5},c,r,l=0,tPM_{2.5}}}|_t}{\partial \overline{E_{NO_x,c,r,l,t}}|_t \overline{E_{TERP,c,r,l,t}}|_t} \overline{E_{NO_x,c,r,l,t}}|_t \overline{E_{TERP,c,r,l,t}}|_t) \tag{17}$$

It would be good to know the specific cause of the instability from a mathematical viewpoint. Can you derive specifically how the instability grows? This is particularly of interest if the hyd code is truly exact as this would seem to imply some level of inexactness.

- Thank you for this question and perceptive comment. ISORROPIA in reverse mode (with aerosol concentration, RH, and temperature as inputs to the model) is called four times in each execution of the aerosol module in CMAQ in the iterative process of estimating the condensation and evaporation of volatile inorganic gases (HCl, HNO$_3$, NH$_3$) to and from coarse-mode inorganic aerosols (e.g., ANO3K, ANH4K) to other modes.

  Prior research has shown that the aerosol thermodynamic model ISORROPIA in reverse mode leads to unrealistic predictions of changes in aerosol pH and H$^+$ concentrations (Hennigan et al., 2015). The H$^+$ concentrations are also extremely sensitive to tiny changes in inputs. This origin of unrealistic sensitivity values is numerically consistent with the underlying equations (based on the ion-balance approach) and, therefore, supports the limits that are based on H$^+$ to OH$^-$ concentrations.

A second constraint is to limit the changes in $H^+$ concentrations among the four ISORROPIA runs. After conducting additional testing, we have observed an exponential growth of sensitivity values during one of the first two calls of the reverse mode of ISORROPIA in the coarse mode hybrid equilibration routine in CMAQ. The use of reverse ISORROPIA in CMAQ is similar to a root-finding based approach, and the initial two calls of reverse ISORROPIA lead to unrealistic $H^+$ concentration changes within a very short timeframe (90 seconds in modeled time). We placed an empirical limit of 1.25 for the changes of $H^+$ concentration through one single reverse ISORROPIA run for one cell. For instance, if the $H^+$ concentration changes by 1.25 times from one to the next run, the changes in sensitivity values are ignored. We were still able to achieve the agreement of first- and second-order sensitivity values with respect to the finite-difference or the hybrid-approach with these limits in place.

The hyd code, by mathematical definitions, gives the numerically exact first- and second-order sensitivities of the variable with respect to the perturbed emission value. The numerically exact nature of the hyperdual approach exposes the shortcomings of reverse ISORROPIA.

One important point to note is that despite this shortcoming of ISORROPIA, the performance of CMAQ for inorganic gases and aerosols does not seem to be widely impacted. It is essential to employ the two constraints mentioned in the main manuscript to stop the exponential growth of sensitivities for now. In the future, better constraints and updates to the original model may be able to eliminate the necessity of such conditions.

We have made the description of the two constraints clearer in lines 219 to 223 of the manuscript.

Table 1: The caption needs to state what is being compared. That information can also go directly on the graphs in Fig. 2, so Table 1 is not needed. It would be more effective that way as well.

- Thank you for the helpful direction. We have augmented the caption for Table 1 as indicated below. Unfortunately, since Fig. 2 is a 12-panel plot, we are concerned that the average reader may not be able to see the correlations unless they are included separately as in Table 1. We have also extended the caption of Table 1 and, similarly, of Table 2.

   **Table 1: The** slope**s** and $R^2$ **values from the linear regression** of the first-order sensitivities of ground layer species concentrations of domain-wide perturbations **by the hyperdual-step method compared to finite difference sensitivities. The gas-phase species sensitivities with respect to their emissions are line three, where APIN denotes $\alpha$-pinene and TERP denotes all other monoterpene species. Line four includes sensitivities of aerosol phase products with respect to their precursors where ANO$_3$ denotes the total aerosol phase nitrate products, ASO$_4$ denotes the total aerosol sulphate products, and $\Sigma$AMT denotes the total aerosol photooxidation products from monoterpene. Line five includes the sensitivities of the total PM$_{2.5}$ concentration with respect to each gas-phase precursor. The visual comparison of the agreement for each relationship is shown** in Figure 2.

   **Table 2: The** slope**s** and $R^2$ **values from the linear regression** of the second-order sensitivities of ground layer species concentrations of domain-wide perturbations **by the**

**hyperdual-step method compared to finite difference sensitivities. The gas-phase species sensitivities with respect to their emissions are line three, where APIN denotes α-pinene and TERP denotes all other monoterpene species. Line four includes sensitivities of aerosol phase products with respect to their precursors where ANO₃ denotes the total aerosol phase nitrate products, ASO₄ denotes the total aerosol sulphate products, and ΣAMT denotes the total aerosol photooxidation products from monoterpene. Line five includes the sensitivities of the total PM$_{2.5}$ concentration with respect to each gas-phase precursor. The visual comparison of the agreement for each relationship is shown** in Figure 5.

Why not compare the results to another sensitivity analysis method implemented in a CTM, e.g., DDM-3D. This would seem to be much more in line with demonstrating the potential advantages of the method.

- At the time of the development and testing, the DDM-3D for CMAQ version 5.3.2 was not developed yet due to the complexity of updates to a DDM-based approach in complex chemical transport models. The development of HDDM in CMAQ was also compared against a finite-difference-based approach (Zhang et al., 2012). Accordingly, such a comparison is the focus of future work and outside the scope of this manuscript.

Given the description of what was involved, it is not apparent how much of a re-coding savings are involved between the hyd approach and others. Maybe a bit more on the relative effort with more specifics.

- The simplicity of recoding the HYD approach for updates relies on the fact that developers do not need to consider the details about the actual update of the model and construct sensitivity (DDM-3D) or adjoint equations based on the update. Instead, we could change the newly added variable types from "REAL" to "HYPERDUAL". The sensitivities are calculated line-by-line based on defined hyperdual sensitivities.

  To give the reader the sense of this simplicity, we have added the following clause to the last sentence of the first paragraph in the conclusions.

  Lines 481 to 482: The development process of CMAQ-hyd is also more straightforward than that of other advanced methods **since all that is needed is to change the type of newly declared variables to hyperdual.**

Did they validate or evaluate the hyperdual module? The two words have rather different meanings.

- Thank you for highlighting this important distinction in language. The reason we use the word "validate" to describe how the hyperdual overloading library was assessed is because these equations are accurate to machine precision. We validated the hyperdual module based on a framework developed by previous work on multicomplex numbers (Pellegrini and Russell, 2016), which facilitates comparison of the hyperdual-based sensitivity to the analytical sensitivity for each of the mathematical functions implemented in the overloading library. We have revised and clarified **lines 194 to 196** of the manuscript.

Lines 194 to 196: Before being applied to CMAQ, the operator overloading library was separately **evaluated against analytical derivatives** using a testing framework developed by Pellegrini and Russell (2016).

One of the more interesting findings of the paper is the computational efficiency found in the hyd method applied to CMAQ vs. other applications. The discuss this a bit, but a bit more analysis would be of interest. For example, for the case of four or eight nodes, say, provide the module-by-module ratio of computational times.

- Thank you for your interest in the detailed computational efficiency of each module. We have developed and now include Figure S3, which demonstrates the relative computational efficiency of each module in the SI of the revised manuscript. We have also revised part of the manuscript to highlight the most interesting findings of profiling, including the relative computational time of the chemistry module (Chem) to the aerosol module (AERO) for CMAQ and CMAQ-hyd.
- In the process of updating this analysis, we identified errant underlying data in the first version of Figure 8. The 'Other' category for 1 node, and the MPI_Barrier for 2 nodes have now been corrected, and we have revised Figure 8.
- We have reorganized and added descriptions of the computational time to make the discussion more comprehensive in **lines 456 to 473**.

Lines 456 to 473: With the same computing resources, the total computation time of the CMAQ-hyd is approximately 2.5 (2.44–2.56) times longer. Despite the additional computation burden, CMAQ-hyd remains computationally competitive with the traditional FDM when calculating derivatives. One run of CMAQ-hyd generates the same amount of first- and second-order sensitivity information as at least three runs of regular CMAQ. The relatively low computational cost of CMAQ-hyd, compared to the previous operator overloading approach, may be due to the selective modification of the source code. In contrast to GEOS-CHEM CVM (Constantin and Barrett, 2014), only parts of the model that involve calculating the main species concentration array use hyperdual calculations.

**The computational time of scientific modules in CMAQ-hyd generally scales well with increases in computational resources, similar to the original CMAQ.** Chem, Aero, and Vdiff are the most computationally expensive modules in both CMAQ and CMAQ-hyd. The relative computational cost of Aero is higher in the CMAQ-hyd than in the regular CMAQ. **The ratio of computational time of Chem to Aero is 1.53 (1.49–1.56) for the CMAQ-hyd runs and 3.98 (3.85–4.19) for the regular CMAQ runs (Fig. 8).** Future work can potentially reduce the computational cost by ignoring sensitivity propagations during the iterative root-finding process in select subroutines, since only the output concentrations from these subroutines are used in the later part of the model. This is also a significant advantage of any operator overloading-based approach (Fike and Alonso, 2011). **The computational time of each module is detailed in Table S3, and the full relative percentage of computational time of each module of eight runs is shown in Figure S3.**

**The MPI_Barrier function also scales well with an increasing number of processors. To a certain point, subdividing the domain further reduces the variability of the time required for science processes to be completed across different nodes, resulting in a reduction of the amount of time the program spends waiting for all processes to be synchronized. One important thing to note here is that the scaling of the MPI_barrier is dependent on the number of nodes to number of grid cells.**

The I/O process of newly added first- and second-order sensitivity output files increases the computational cost; however, the I/O of species concentration files has a much lower computational cost than other computing modules in CMAQ for this specific scenario. **The I/O processes of CMAQ-hyd and CMAQ take 193 (181–206) seconds and 52 (47–56) seconds, respectively. The I/O process in CMAQ-hyd takes approximately 3.7 times longer on average than that in the regular CMAQ. The overall memory overhead of the CMAQ-hyd is approximately 25 GB for this simulation. A parallel input/output (I/O) approach may be applied to reduce the possibility of potential memory overflow in processor 0 (Wong et al., 2015).**

References

Fike, J. and Alonso, J.: The Development of Hyper-Dual Numbers for Exact Second-Derivative Calculations, 49th AIAA Aerospace Sciences Meeting including the New Horizons Forum and Aerospace Exposition, 2011-01-04, 10.2514/6.2011-886,

Hennigan, C. J., Izumi, J., Sullivan, A. P., Weber, R. J., and Nenes, A.: A critical evaluation of proxy methods used to estimate the acidity of atmospheric particles, Atmospheric Chemistry and Physics, 15, 2775-2790, 10.5194/acp-15-2775-2015, 2015.

Pellegrini, E. and Russell, R. P.: On the Computation and Accuracy of Trajectory State Transition Matrices, Journal of Guidance, Control, and Dynamics, 39, 2485-2499, 10.2514/1.G001920, 2016.

Zhang, W., Capps, S. L., Hu, Y., Nenes, A., Napelenok, S. L., and Russell, A. G.: Development of the high-order decoupled direct method in three dimensions for particulate matter: enabling advanced sensitivity analysis in air quality models, Geoscientific Model Development, 5, 355-368, 10.5194/gmd-5-355-2012, 2012.

---

## Author Comment (AC3)

**Response to RC3**

Jiachen Liu[1], Eric Chen[1], Shannon L. Capps[1]

[1]Department of Civil, Architectural & Environmental Engineering, Drexel University, Philadelphia, Pennsylvania, USA

*Correspondence to*: Shannon L. Capps (shannon.capps@drexel.edu)

*For the convenience of the reviewer, we have included the comments from Reviewer 3 in black, normal font style and our responses indented in green, normal font style. The changes made to the manuscript or supplemental information corresponding to our response are provided in green, bold style font with the context of line numbers from the original manuscript and the original, remaining text in the normal font style.*

This manuscript describes the theoretical base and implementation of the new CMAQ-hyd model for calculation of the first and second order sensitivities. This model is based on the hyperdual step method implemented into the widely used CTM model CMAQ (US-EPA). The manuscript describes also the evaluation and testing of the model including performance tests. The work brings significant new scientific results in the area of air quality modelling. The hyperdual step method has been already known but its utilisation in the CTM models is novel and very beneficial for the scientific community. The paper is well structured and clearly written. The findings are generally well described, I have a few specific comments described below. All corresponding materials (model source code and testing data) are available which allow to reproduce the experiments. The topic of the paper fits very well to GMD scope and I recommend the manuscript for publication in GMD after a minor revision (see specific comments below).

- We would like to thank the reviewer for their favorable and helpful comments on our manuscript. We have addressed the specific comments that the reviewer provided and consider the manuscript to be strengthened through this revision.

Specific comments:

I. Comments to the manuscript:

l. 149, Abbreviation SI is not defined. You probably meant Supplementary Material. Please, use the full name in this first occurrence of the abbreviation.

- Thank you for the correction. We have revised the manuscript to mention the first occurrence of Supplemental Information (SI).

    Lines 149 to 150:

    A demonstration of several basic operations is provided in the **Supplemental Information (SI)** while a more detailed discussion of the mathematical properties of hyperdual numbers is given by Fike and Alonso (2011).

l. 151-153, 162: The relation of a1, a2, a12 and Hh is not clearly formulated. Please, try to reformulate to make this step more comprehensible to the reader.

- Thank you for highlighting the need for more clarification on applying the hyperdual step method. Reviewer 1 expressed similar concerns. To address both comments, we revised the

description of the multiplicative hyperdual perturbation in the text and the SI. The changes to Lines 151 to 159 and to Line 15 in the SI are shown below.

Lines 151-159:

**Akin to the Taylor series expansion about the real value of $x$ in the finite difference method**, the method of ascertaining sensitivities through a perturbation in **hyperdual space is based on a Taylor series expansion in an orthogonal dimension of the number. Specifically,** a hyperdual number with unity in $a_0$ and unity in one in $a_1$ and $a_2$, **is multiplied** with the independent variable of interest before operating on it. After model execution, a Taylor series expansion is applied to extract sensitivities. **For instance,** the hyperdual-step method is applied to a scalar function, $f(x)$, by multiplying $x$ by a hyperdual number, $H_h$, where $H_h = 1.0 + h_1\epsilon_1 + h_2\epsilon_2$. **This product** results in:

$$
\begin{aligned}
f(xH_h) = f(x) &+ (xh_1\epsilon_1 + xh_2\epsilon_2)f'(x) \\
&+ \frac{1}{2!}(xh_1\epsilon_1 + xh_2\epsilon_2)^2 f''(x) \\
&+ \frac{1}{3!}(xh_1\epsilon_1 + xh_2\epsilon_2)^3 f'''(x) + \cdots
\end{aligned}
\tag{5}
$$

where "…" represents higher order terms in the series. Eliminating all terms that are zero due to the definition of hyperdual numbers (Eq. 2) **leads to**

$$
f(xH_h) = f(x) + (xh_1\epsilon_1 + xh_2\epsilon_2)f'(x) + x^2 h_1 h_2 \epsilon_{12} f''(x)
\tag{6}
$$

**where $f(xH_h)$ is a hyperdual number.**

The properties of hyperdual numbers (Eqs. 2–4) lead to two significant results. First, all terms in the Taylor series expansion with derivatives higher than second-order become zero because all values include $\epsilon_1^2$, $\epsilon_2^2$, or $\epsilon_{12}^2$. Second, the real component is unchanged. **A more detailed expansion of terms can be found in Eq. S7 in the SI or in the original development of hyperdual numbers, following the multiplication rule between a hyperdual and a real number** (Fike and Alonso, 2011).

Line 15, SI:

$$
\begin{aligned}
f(xH_h) &= f\big(x(1.0 + h_1\epsilon_1 + h_2\epsilon_2)\big) \\
&= f(x + xh_1\epsilon_1 + xh_2\epsilon_2) \\
&= f(x) + (xh_1\epsilon_1 + xh_2\epsilon_2)f'(x) + \frac{1}{2!}(xh_1\epsilon_1 + xh_2\epsilon_2)^2 f''(x) + \frac{1}{3!}(xh_1\epsilon_1 + xh_2\epsilon_2)^3 f'''(x) + \cdots \\
&= f(x) + (xh_1\epsilon_1 + xh_2\epsilon_2)f'(x) + \frac{1}{2}\big(x^2 h_1^2 \epsilon_1^2 + 2x^2 h_1\epsilon_1 h_2\epsilon_2 + x^2 h_2^2 \epsilon_2^2\big)f''(x) \\
&= f(x) + xh_1\epsilon_1 f'(x) + xh_2\epsilon_2 f'(x) + x^2 h_1 h_2 \epsilon_{12} f''(x)
\end{aligned}
$$

**The Taylor series expansion of the multiplicative hyperdual perturbation (Eq. 5) is shown in Eq. S7.**

l. 214-232: This paragraph is not well formulated and the explanations are slightly disarranged. E.g. utilisation of the forward and reverse mode in CMAQ is not explained clearly. Similarly, the

adjustments done in the CMAQ itself, CMAQ-adjoint, and CMAQ-hyd are not clearly distinguished and all paragraph needs multiple readings to comprehend meaning of it. Please, try to reformulate it in a more straightforward way to allow easy understanding also to reader which is not fully familiar with the details of the CMAQ internals.

- Thank you for pointing out the need to rearrange explanations from Line 214 to 232. We have reformulated the paragraph to make it more logical.

  Lines 214-232:

  **Several** source code alternations were made to reduce the complexity of development and overcome the numerical instabilities related to hyperdual calculations in CMAQ's treatment of aerosol **specifically within the inorganic thermodynamic module ISORROPIA (Fountoukis and Nenes, 2007; Nenes et al., 1998). For the simplicity of development, we applied a Fortran 90-compliant version of ISORROPIA to replace the original Fortran 77 version of ISORROPIA in CMAQ.**

  **ISORROPIA, as a key component of the aerosol module in CMAQ, is called either in** the forward or the reverse mode. The forward mode of ISORROPIA takes the sum of gas and aerosol species concentrations, along with the relative humidity and temperature, to determine the partitioning of **Aitken- and accumulation-mode** species across the gas and aerosol phases **in CMAQ. In the original CMAQ model, ISORROPIA is run in the forward mode without limiting the temperature and pressure of the simulation. The determination process of species concentrations involves an iterative method which sometimes is numerically unstable during iterations for upper layer cells with low temperature and pressure for sensitivity computations with the HYD. To increase the numerical stability of CMAQ-hyd, we implemented temperature and pressure constraints so that the forward-mode ISORROPIA is only called when the cell temperature exceeds 260 K, and cell pressure exceeds 20,000 Pa. A similar set of temperature and pressure limits was applied to the call of ISORROPIA in the adjoint of CMAQ (Zhao et al., 2020). These changes do not affect the species concentrations computed by CMAQ while ensuring that the sensitivity computation process is stable.**

  **To calculate the dynamic equilibrium of coarse mode aerosol species with the gas phase (Pilinis et al., 2000; P. Capaldo et al., 2000), CMAQ employs** the reverse mode of ISORROPIA. **The input to reverse mode ISORROPIA includes concentrations of aerosol species, relative humidity, and temperature, and it results in partitioned concentrations in the** solid, liquid, and gas phases. The reverse-mode solution leads to unrealistic sensitivities calculated by HYD when the aerosol pH is close to neutral. One previous study found that the reverse ISORROPIA fails to capture the actual behaviour of inorganic aerosol when the pH is close to 7 (Hennigan et al., 2015). **To ensure stability of the sensitivity calculations, the changes to the hyperdual components in the coarse mode dynamic equilibrium are ignored** when the pH of coarse mode aerosol is close to neutral, **which ensures that the real components are identical to the original model.**

l. 242-243: The definition of Cpm2.5 and Enox should be moved forward somewhere to line 238 before their utilisation.

- Thank you for noticing the mentioning of Cpm2.5 and Enox before they are formally defined. We have rearranged the paragraph. To address the comment of Reviewer #1, we have also made some additional changes to the paragraph.

Lines 236-250:

Here, **for the sake of illustration**, we consider the semi-normalised sensitivities of **time-averaged** output concentrations of **ground-level** PM$_{2.5}$ concentrations, $C_{PM_{2.5},c,r,l=0,t}$, to input NO$_x$ (NO+NO$_2$) emissions, $E_{NO_x,c,r,l,t}$, averaged over time, **$t$, for any given cell as indicated by the column, $c$, and row, $r$.** First-order semi-normalised sensitivities, $s_{NO_x}^{PM_{2.5}}$, and second-order semi-normalised sensitivities, $s_{NO_x}^{(2)PM_{2.5}}$ exemplify sensitivities relevant to environmental decision makers (Eqs. **15–16**).

Semi-normalised sensitivities reduce the complexity of interpretation by providing sensitivities in the units of the concentration per percent change of emissions. The semi-normalised sensitivities also scale down the impact from cells with low emission rates, which is consistent with the concentration reduction that is realistic **to expect**. Similarly, the time-averaged, semi-normalised cross-sensitivity of PM$_{2.5}$ to both NO$_x$ and monoterpene is denoted as $s_{NO_x,TERP}^{(2)PM_{2.5}}$, with $E_{TERP}$ representing the emission of monoterpenes (Eq. **17**).

$$s_{NO_x,TERP}^{(2)PM_{2.5}} = \frac{\partial^2 \overline{C_{PM_{2.5},c,r,l=0,t}PM_{2.5}}|_t}{\partial \overline{E_{NO_x,c,r,l,t}}|_t \overline{E_{TERP,c,r,l,t}}|_t} \overline{E_{NO_x,c,r,l,t}}|_t \overline{E_{TERP,c,r,l,t}}|_t) \tag{17}$$

l. 252: Superscripts inc, dec, and orig - explicit description of the meaning of these abbreviations might be beneficial even the following example gives a hint.

- Thank you for your suggestion. We have added an additional description of the meaning of these abbreviations as shown below.

Lines 251-252:
where the subscripts $c$, $r$, and $l$ represent the column, row, and layer; the subscript $t$ represents the time from the start of the model run; and the superscripts $inc$, $dec$, and $orig$ represent the initial perturbation direction **(i.e., increased, decreased, and original emissions, respectively)**.

l.261-262: The last sentence partially repeats the statement of the sentence on lines 257-258. Moreover, section 2.1 discusses the hyperdual method and its errors while errors of the central FDM are discussed in the section 1 (l. 69-88).

- Thank you for pointing out the repetition of statements in these lines. We have removed this sentence from the manuscript.

l.269-270: The formulation "The second-order sensitivity evaluation is between a hybrid hyperdual-finite-difference method (HYD-FDM) and the hyperdual-step method." seems to be unclear. The approach is explained in the following text but the sentence can confuse the reader at the beginning. Please, reformulate.

- Thank you for pointing out the confusing of statements in these lines. We have reformulated the sentence as shown below.

Lines 268-271:

Although the FDM can be applied to compute second-order sensitivities in CMAQ, previous studies have shown that the results are noisy and highly dependent on the perturbation sizes (Zhao et al., 2020; Zhang et al., 2012). **In order to evaluate the second-order sensitivities computed by the HYD method, we adopted a hybrid hyperdual-finite-difference method (HYD-FDM).** The **HYD-FDM** sensitivity calculation is given by:

l.286-287: You state here the evaluation has been done with 50% perturbation but in l.254-255 you assert the perturbation used has been 125% and 75%, i.e. 25%. I may overlook something and it may represent a different perturbation. Please, either correct these numbers (in case of the mistake) or add better explanation or description of these numbers (in case they are correct from some reason).

- Thank you for your suggestion. We have applied the central difference method with perturbations in both the increasing (125%) and decreasing directions (75%). We have added the following line to better explain the perturbation for finite difference.

  Line 285-289:
  We evaluated the implementation of CMAQ-hyd by comparing the first-order sensitivities of various species in CMAQ calculated by HYD with a hyperdual-step perturbation described in Section 2.3 (HYD sensitivities) and FDM with a domain-wide emission perturbation (FD sensitivities). **The FD sensitivities were computed with the difference between a 25% increase and a 25% decrease in domain-wide emissions.** Overall, different HYD and FD sensitivities agree well, as evidenced by the close alignment of the points on the blue identity line, which represents perfect agreement, in most panels of Figure 2.

l. 293-316: The Fig. 3 seems to be poorly arranged as the overlapping points do not allow the comparison of individual results. My suggestion is to rearrange the Fig. 3 in a way which will allow to study better the behaviour of the FDM subtraction and truncation errors with decreasing of the perturbations and to assess the "convergence" of FDM to hyperdual results. Four separate graphs might work better than the current unified graph. You can have a better idea how to deal with it.

- Thank you for your suggestion about rearranging Figure 3 to make the description clearer. We have altered Figure 3 so that it depicts the four separate approaches (125%, 75% FDM; 110%, 90% FDM; 105%, 95% FDM; HYD) in the modelling domain on a map. We have removed the forward and backward finite differences (125%, 100%; 100%, 75%) and added an additional central-difference-based calculation (105%, 95%).
- From the maps in the new Figure 3, we can observe that the FDM does not "converge" to the hyperdual results when we decrease the perturbation sizes. Due to numerical noise inherent to CMAQ, the sensitivities calculated with runs with smaller perturbation sizes deviate farther from the hyperdual results. This result is also described in a previous work (Zhang et al., 2012).
- We have also changed Figure 4 accordingly to reflect the change in computational scenarios used.
- We have removed the original Figure S1 since it is similar to the new Figure 3 in the main manuscript.

Lines 317 to 349:

   The FD sensitivities with the base case perturbation (125 %, 75 %) and **two** other perturbation size pairs (110 %, 90 %; **105 %, 95 %**) are shown in Fig. 3. The FDM sensitivities calculated with different perturbation sizes **are plotted on Fig. 3a, Fig. 3b, and Fig 3c., respectively. The FDM sensitivities exhibit similar behaviour to the HYD sensitivities over the continents. However, the** inconsistency among the sensitivities calculated by FDM with different perturbation sizes **over the ocean** (Fig. 3) suggests that the FD sensitivities heavily depend on the perturbation sizes. This result **demonstrates** the **relatively** low credibility of FD sensitivities, particularly for highly nonlinear relationships where the truncation errors could be large. **Notably, reducing perturbation sizes in the FDM did not lead to convergence with hyperdual sensitivities. This divergence may be attributed to the propagation of numerical noise from the model run to the calculated sensitivities as perturbation sizes decrease. This finding is consistent with the results in Zhang et al. (2012).** Our findings demonstrate the importance of using other methods, including the HYD, which are not prone to truncation or cancellation errors for **probing** nonlinear relationships in CTMs.

   We also compared the spatial distribution of HYD sensitivities (Fig. 4a) against the average (Fig. 4b) and the range (Fig. 4c) of the FD sensitivities with **three** different perturbation sizes. Differences are evident between the HYD and the average FD sensitivities in central North Carolina and Tennessee as well as off the coasts of Georgia and South Carolina. The HYD predicts slightly negative sensitivities in North Carolina and Tennessee while the FDM predicts slightly positive values. The average FDM sensitivities off the coast of Georgia and South Carolina were noisy, with alternating positive and negative sensitivities, while the HYD sensitivities were much less noisy.

L. 393-394: According figures 6g and 6h, the second order sensitivities of PM2.5 to TERP (Fig. 6g) are mostly negative, while to APIN (Fig. 6h) are mostly positive.

- Thank you for pointing out the mistake in the manuscript. We have corrected the mistake in lines 393-394.

   Lines 393-394:
   On the other hand, the $s_{TERP}^{(2)PM_{2.5}}$ (Fig. 6g) is mostly **negative**, while $s_{APIN}^{(2)PM_{2.5}}$ (Fig. 6h) is mostly **positive**.

l. 445-446: The implementation of CVM is not higher. You probably meant the computational cost of this CVM based model.

- Thank you for pointing out the mistake in the manuscript. We have corrected the mistake in lines 445-446.

   Lines 445-446:
   For instance, the implementation of CVM in GEOS-CHEM **results in** a 4.5-**fold increase in computational overhead when compared to** the **standard** model (Constantin and Barrett, 2014).

l. 448-449: The sentence does not make sense. Probably words were left out after "are shown for the".

- Thank you for pointing out the unclear sentence in the manuscript. We have rewritten the sentence to make it clearer.

  Lines 448-449:
  The total wall time **with different numbers of computational nodes used** for identical runs of the original CMAQ model and CMAQ-hyd **is displayed** (Fig. 8).

l. 450: You hide from the reader the number of processor cores/MPI processes. This fact is much more important than number of nodes. The 7rganization of the MPI processes to individual nodes can only influence the Infiniband (or another transport layer) overhead which is usually small in modern HPC system for such a type of tasks. Also, the extent of your configuration is important mainly for assessment of the parallelization efficiency with growing number of MPI processes involved as well as for memory demands of the model. Please, give the reader full information about your testing configuration. You can add this information into the Section 4. Of Supplements and give a reference to it here.

- Thank you so much for your request of the detailed configuration of computing nodes. We have provided the additional information in the SI and added a sentence in the main manuscript referring the readers to the SI.

  Line 450:
  The CMAQ-hyd and regular CMAQ runs were performed with 1, 2, 4, and 8 nodes on the supercomputing cluster. **The configuration of computing resources is detailed in Section 4 of the SI.**

  Lines 42-45, SI: **The runs using 1, 2, 4, and 8 nodes runs use 36 processors, 72 processors, 144 processors, and 288 processors, respectively. CMAQ delegates the computing job of the grid by partitioning in the horizontal plane. Each processor is tasked with executing the scientific processes for a set of columns before communicating the select information on the boundaries. One example of the subdivision of the horizontal plane by 1 node (36 processors) is shown in Figure S3. Figure S3a demonstrates the 100 by 80 horizontal grid of our modelling domain, and Figure S3b demonstrates the subdivision of computing tasks by 36 processors. Each processor is responsible for computations in rectangular squares drawn in dashed lines. The columns of CMAQ are decomposed into 36 MPI processes in the shape of 6 columns and 6 rows of MPI processes. The specific shape of MPI processes is: 6 columns by 6 rows for 1 node (36 processors), 12 columns by 6 rows for 2 nodes (72 processors), 12 columns by 12 rows for 4 nodes (144 processors), and 24 columns by 12 rows for 8 nodes (288 processors). We applied the same computational configuration for the regular CMAQ and the CMAQ-hyd runs during code profiling processes.**
  **When the number of processors used increases, the number of grid cells along the boundaries requiring information exchange increases. Consequently, there is an increase in relative time consumed by the I/O processes when the number of processors used increases.**

l.483: I have slight doubts you can use the word "validated" here, I would suggest "evaluated" or a similar word. My reasoning is methodological. The purpose of tests done in section 3.1 is to evaluate the correctness and accuracy of the hyperdual based implementation. You compare results of the new more precise method with an established less precise method (according the theory) and you get some differences. You attribute these discrepancies between FDM and hyperdual results only to the nonlinearity of the model but how can you be sure they are not caused also by another reason, e.g. some problem in the hyperdual implementation? Yes, I am also convinced, that it is the result of FDM errors due to nonlinearity of the model but it is not a formal proof. You give a good supporting arguments in the following parts of the section 3.1 and they very well support the trust in the correctness of the CMAQ-hyd model implementation. But I still would be careful to call it formally validation. The thorough evaluation of the behaviour of CMAQ-hyd and its comparison with FDM done in 3.1 shows that the differences are of expected properties what allows to trust this new model.

- Thank you so much for this valuable input. We validated the hyperdual calculations defined in the HDMod operator overloading library. Motivated by a similar request for clarity in this comment and another of Reviewer #1, we have added a sentence to Lines 215-216 to explain the validation against analytical derivatives with a testing framework developed by Pellegrini and Russell (2016).
- Indeed, the comparison between the hyperdual-step method and the finite difference method does not serve as a formal proof. The word 'evaluated' is a good fit for describing the relationship between sensitivities calculated by CMAQ with finite difference and CMAQ-hyd.

  Lines 215-216:
  Before being applied to CMAQ, the operator overloading library was separately **validated by comparing against analytical derivatives using a testing framework developed by Pellegrini and Russell (2016).**

  Line 483-486:
  We developed and validated the hyperdual-step module "HDMod", which **is limited to analytically verifiable** mathematical operations **of hyperdual numbers**. This module can also be applied to other numerical models where first- and second-order sensitivities are of interest. We further **evaluated** the development of CMAQ-hyd against the FDM and FDM-HYD hybrid method to ensure the correctness of the implementation. During the **evaluation** process, …

l. 507: "..free from numerical noise.." - Only specific types of the numerical errors (truncation and subtractive cancellation errors) are eliminated by this method. Even they are the most important ones, I would suggest a more careful formulation here.

- Thank you for suggesting the specific types of the numerical errors that the hyperdual-step method can avoid. We have updated the manuscript accordingly.

  Lines 486-487: CMAQ-hyd **computes** sensitivities free from **truncation and subtractive cancellation errors**, **unlike** those calculated by the FDM.

Lines 506-507: In conclusion, we have developed and evaluated CMAQ-hyd, a novel, augmented model to compute first-order, second-order, and cross-sensitivities free from **subtractive cancellation and truncation errors** in CMAQ.

References

Constantin, B. V. and Barrett, S. R. H.: Application of the complex step method to chemistry-transport modeling, Atmospheric Environment, 99, 457-465, https://doi.org/10.1016/j.atmosenv.2014.10.017, 2014.

Fike, J. and Alonso, J.: The Development of Hyper-Dual Numbers for Exact Second-Derivative Calculations, 49th AIAA Aerospace Sciences Meeting including the New Horizons Forum and Aerospace Exposition, 2011-01-04, 10.2514/6.2011-886,

P. Capaldo, K., Pilinis, C., and Pandis, S. N.: A computationally efficient hybrid approach for dynamic gas/aerosol transfer in air quality models, Atmospheric Environment, 34, 3617-3627, https://doi.org/10.1016/S1352-2310(00)00092-3, 2000.

Pellegrini, E. and Russell, R. P.: On the Computation and Accuracy of Trajectory State Transition Matrices, Journal of Guidance, Control, and Dynamics, 39, 2485-2499, 10.2514/1.G001920, 2016.

Pilinis, C., Capaldo, K. P., Nenes, A., and Pandis, S. N.: MADM-A New Multicomponent Aerosol Dynamics Model, Aerosol Science and Technology, 32, 482-502, 10.1080/027868200303597, 2000.

Zhang, W., Capps, S. L., Hu, Y., Nenes, A., Napelenok, S. L., and Russell, A. G.: Development of the high-order decoupled direct method in three dimensions for particulate matter: enabling advanced sensitivity analysis in air quality models, Geoscientific Model Development, 5, 355-368, 10.5194/gmd-5-355-2012, 2012.

Zhao, S., Russell, M. G., Hakami, A., Capps, S. L., Turner, M. D., Henze, D. K., Percell, P. B., Resler, J., Shen, H., Russell, A. G., Nenes, A., Pappin, A. J., Napelenok, S. L., Bash, J. O., Fahey, K. M., Carmichael, G. R., Stanier, C. O., and Chai, T.: A multiphase CMAQ version 5.0 adjoint, Geoscientific Model Development, 13, 2925-2944, 10.5194/gmd-13-2925-2020, 2020.

---

## Author Comment (AC4)

**Response to RC3**

Jiachen Liu[1], Eric Chen[1], Shannon L. Capps[1]

[1]Department of Civil, Architectural & Environmental Engineering, Drexel University, Philadelphia, Pennsylvania, USA

*Correspondence to*: Shannon L. Capps (shannon.capps@drexel.edu)

*For the convenience of the reviewer, we have included the comments from Reviewer 3 in black, normal font style and our responses indented in green, normal font style. The changes made to the manuscript or supplemental information corresponding to our response are provided in green, bold style font with the context of line numbers from the original manuscript and the original, remaining text in the normal font style.*

II. Comments to the model source code:

1. The source files of the HDMod and HDMod_cplx do not contain license, authors, and reference to original author of the C++ module which it is based on. Please, fill it in.

- Thank you for pointing out the lack of detailed documentation of license, authors, and reference to the original author of the C++ module. We have updated the HDMod and HDMod_cplx files to include the license, authors, and reference to the original C++ module.

Lines 513-514:
The CMAQ-hyd model is archived at https://doi.org/10.5281/zenodo.**10119026** (Liu et al., 2023).

2. The changes in the source code contain almost no comments. It would be beneficial to denote and comment your adaptations as it could help users to quickly orient in your changes and it also would be of use during adaptation of the CMAQ-hyd to the new versions of the CMAQ model. I do not insist on doing it before the publication of this paper but I consider beneficial to do it as soon as possible.

- Thank you for your suggestion about adding comments to the source code. Since we submitted the paper, we have made several updates to the source code and plan to add comments to our adaptations in the near future.

References

Liu, J. C., Eric, Capps, Shannon: CMAQv5.3.2-hyd (5.3.2-hyd1.0.1), Zenodo [code], 10.5281/zenodo.10119026, 2023.